# Biallelic mutations in *calcium release activated channel regulator 2A (CRACR2A)* cause a primary immunodeficiency disorder

Beibei Wu[1†], Laura Rice[2†], Jennifer Shrimpton[3], Dylan Lawless[4], Kieran Walker[3], Clive Carter[5], Lynn McKeown[6], Rashida Anwar[2], Gina M Doody[3], Sonal Srikanth[1], Yousang Gwack[1*‡], Sinisa Savic[5,7*‡]

[1]Department of Physiology, David Geffen School of Medicine, UCLA, Los Angeles, United States; [2]Leeds Institute of Medical Research, University of Leeds, Leeds, United Kingdom; [3]Division of Haematology and Immunology, Leeds Institute of Medical Research, University of Leeds, Leeds, United Kingdom; [4]Global Health Institute, School of Life Sciences, École Polytechnique Fédérale de Lausanne, Lausanne, Switzerland; [5]Department of Clinical Immunology and Allergy, St James's University Hospital, Leeds, United Kingdom; [6]Leeds Institute of Cardiovascular and Metabolic Medicine, University of Leeds, Leeds, United Kingdom; [7]National Institute for Health Research-Leeds Biomedical Research Centre and Leeds Institute of Rheumatic and Musculoskeletal Medicine, Wellcome Trust Brenner Building, St James's University Hospital, Leeds, United Kingdom

*For correspondence:
ygwack@mednet.ucla.edu (YG);
S.Savic@leeds.ac.uk (SS)

†These authors contributed equally to this work
‡These authors also contributed equally to this work

**Abstract** CRAC channel regulator 2 A (CRACR2A) is a large Rab GTPase that is expressed abundantly in T cells and acts as a signal transmitter between T cell receptor stimulation and activation of the $Ca^{2+}$-NFAT and JNK-AP1 pathways. CRACR2A has been linked to human diseases in numerous genome-wide association studies, however, to date no patient with damaging variants in CRACR2A has been identified. In this study, we describe a patient harboring biallelic variants in *CRACR2A* [paternal allele c.834 gaG> gaT (p.E278D) and maternal alelle c.430 Aga > Gga (p.R144G) c.898 Gag> Tag (p.E300*)], the gene encoding CRACR2A. The 33-year-old patient of East-Asian origin exhibited late onset combined immunodeficiency characterised by recurrent chest infections, panhypogammaglobulinemia and CD4+ T cell lymphopenia. In vitro exposure of patient B cells to a T-dependent stimulus resulted in normal generation of antibody-secreting cells, however the patient's T cells showed pronounced reduction in CRACR2A protein levels and reduced proximal TCR signaling, including dampened SOCE and reduced JNK phosphorylation, that contributed to a defect in proliferation and cytokine production. Expression of individual allelic mutants in CRACR2A-deleted T cells showed that the CRACR2A[E278D] mutant did not affect JNK phosphorylation, but impaired SOCE which resulted in reduced cytokine production. The truncated double mutant CRACR2A[R144G/E300*] showed a pronounced defect in JNK phosphorylation as well as SOCE and strong impairment in cytokine production. Thus, we have identified variants in *CRACR2A* that led to late-stage combined immunodeficiency characterized by loss of function in T cells.

## Editor's evaluation

The study is the first report and characterization of impaired immune function in an individual with biallelic mutations in the CRACR2A gene. The mechanistic insights from this study, are an important

advance in the understanding of CRAC channels and the regulation of calcium dynamics in the T-cell lineage. The work will be of interest to cell biologists, immunologists and those with interests in intracellular signaling.

## Introduction

T cell activation following antigen recognition involves several signaling events. Amongst others, these include $Ca^{2+}$ influx and activation of the c-Jun N-terminal kinase (JNK) and p38 mitogen-activated protein kinase (MAPK) pathways. The $Ca^{2+}$ influx is initiated shorty after T cell receptor (TCR) engagement, and involves activation of phospholipase $C\gamma$ and production of inositol-1,4,5-trisphosphate ($IP_3$). The latter binds to $IP_3$ receptors in the endoplasmic reticulum (ER) membrane, leading to receptor-mediated release of $Ca^{2+}$ from the ER. The $Ca^{2+}$ release is detected by stromal interaction molecules (STIM) - STIM1 and STIM2, which are EF-hand-containing proteins, located in the ER membrane (*Roos et al., 2005*; *Liou et al., 2005*). Depletion of the ER $Ca^{2+}$ stores leads to $Ca^{2+}$-loss from the STIM1 EF-hand causing it to multimerize and translocate to the ER in close proximity to the plasma membrane (PM). Subsequently, STIM1 binds to PM-resident ORAI1, which is the pore subunit of the $Ca^{2+}$ release–activated $Ca^{2+}$ (CRAC) channel, resulting in the opening of the channel and $Ca^{2+}$ influx (*Feske et al., 2006*; *Zhang et al., 2006*; *Vig et al., 2006*). This process is otherwise known as store-operated $Ca^{2+}$ entry (SOCE) since it depends on the filling state of the ER $Ca^{2+}$ stores. The importance of $Ca^{2+}$ influx for lymphocyte proliferation, differentiation and function, and the role that STIM1 and ORAI1 play in these processes, is illustrated by rare primary immunodeficiency syndromes caused by autosomal recessive (AR) loss of function (LOF) or null mutations in *ORAI1* and *STIM1* genes. The affected patients tend to present from early age with varying degrees of immune defects, ranging from severe combined immunodeficiency (SCID), to milder forms, where the immune impairment is functionally evident, but not associated with any apparent immunodeficiency (*Feske et al., 2006*; *Chou et al., 2015*; *Fuchs et al., 2012*; *McCarl et al., 2009*; *Picard et al., 2009*; *Parry et al., 2016*; *Rice et al., 2019*; *Byun et al., 2010*; *Schaballie et al., 2015*). Patients with SCID-like phenotype present with severe viral, bacterial, and fungal infections. In many cases there is associated immune dysregulation manifesting as lymphoproliferation or autoimmunity (*Picard et al., 2009*). In most cases it is the function rather than numbers of lymphocytes that is affected. Lastly, non-immunological manifestations are common, such as congenital muscular hypotonia, defects in dental enamel development, and anhidrosis, due to critical role of SOCE for these physiological functions (*Parry et al., 2016*; *Rice et al., 2019*; *Wang et al., 2014*; *Lian et al., 2018*).

CRAC Channel Regulator 2 A (CRACR2A) was identified as a protein that binds ORAI1/STIM1 and stabilizes their interaction to mediate SOCE in T cells (*Srikanth et al., 2010a*). Two isoforms of the protein are expressed in T cells, a short cytoplasmic isoform CRACR2A-c, that was originally shown to bind ORAI1/STIM1 and stabilize their interaction, and a longer isoform CRACR2A-a (hereafter referred to as CRACR2A), that is a Rab GTPase. CRACR2A (also called EFCAB4B and Rab46) is abundantly expressed in T cells and belongs to a family of large Rab GTPases that also includes Rab44 and Rab45. It is a protein with multiple functional domains including an N-terminal $Ca^{2+}$-binding EF-hand (also present in the shorter CRACR2A-c isoform), a protein-interacting coiled coil domain and proline-rich domain, and a C-terminal Rab GTPase domain. This long isoform is prenylated at its C terminus and predominantly localizes to vesicles and Golgi in T cells. Both the isoforms interact with ORAI1 and STIM1 and are involved in regulation of SOCE (*Srikanth et al., 2010a*). In addition to regulation of SOCE, CRACR2A has also been shown to interact with Vav1 and regulate JNK phosphorylation. The GTPase domain of CRACR2A and prenylation have been shown to be necessary for its role in JNK activation (*Srikanth et al., 2016*).

The human *CRACR2A* (formerly *EFCAB4B*) gene is located on chromosome 12. Associations between polymorphisms in CRACR2A and human diseases have been identified from numerous genome-wide association studies (GWAS) of Parkinson disease, nonalcoholic fatty liver disease, atrial fibrillation, and chronic infection of HIV type (*Fung et al., 2006*; *Chalasani and Björnsson, 2010*; *Tan et al., 2013*; *Iglesias-Ussel et al., 2013*; *Edelman et al., 2015*). However, until now, CRACR2A has not directly been linked with a primary immunodeficiency (PID) in humans. Here, we describe a single case of late onset combined immunodeficiency due to biallelic variants in *CRACR2A* which encodes CRACR2A. One of the alleles encodes a truncated protein containing another point mutation

in the EF-hand domain (CRACR2A p.[R144G/E300*], hereafter referred to as double mutant DM), whereas the other allele encodes a point mutant (CRACR2A p.E278D), with the mutation residing in the protein-interacting coiled coil region of the protein. Exogenous expression of individual allelic mutations in T cells shows that the DM impairs both SOCE as well as JNK phosphorylation, whereas p.E278D selectively impairs SOCE.

## Results

### Case presentation and immunological work up

We evaluated a 33-year-old male of East-Asian origin. He originally presented to gastroenterology services at the age of 19 years with chronic diarrhea. Following initial assessment, which included normal colonoscopy and negative coeliac screen, he was found to have severe panhypogammaglob-ulinaemia and was referred to clinical immunology for further investigations. On further questioning the patient reported a 5-year long history of recurrent lower respiratory tract infections requiring treatment with antibiotics. He also reported having a left lower lobe pneumonia at 15 years of age. He had no other significant infection history and reported being generally well during his childhood. His only other past medical history of note was appendicectomy at the age of 10 years. He had no

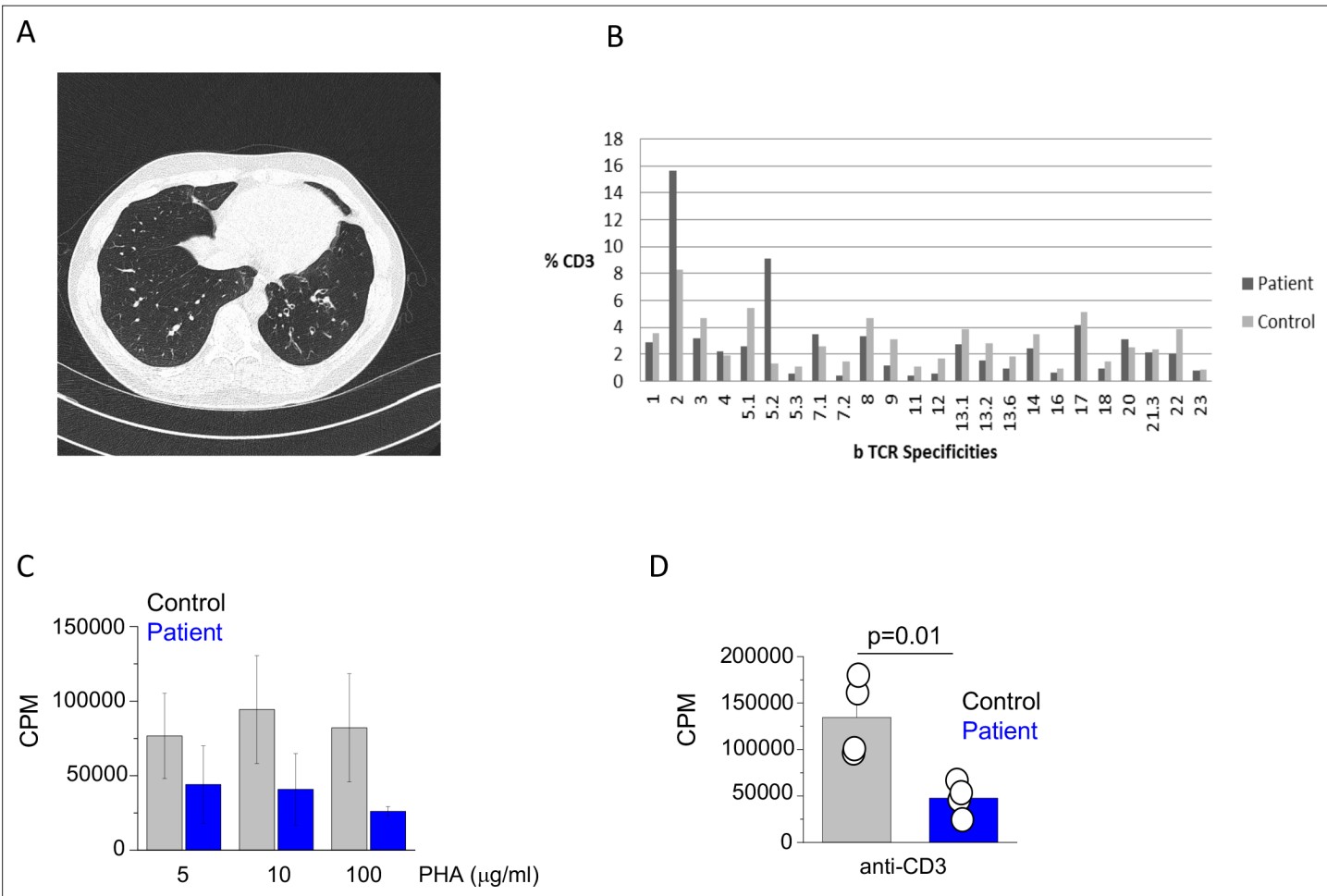

**Figure 1.** Clinical and immunological features of index patient. (**A**) A computerized tomography scan of the chest showing area of cylindrical bronchiectasis at the basal segment of the left lower lobe. (**B**) TCR repertoire as assessed by T receptor spectraphenotyping. Data are representative of one independent assay. (**C**) Phytohaemagglutinin (PHA) and (**D**) anti-CD3 T cell proliferation. CPM (counts per minute). Data are representative of three independent assays.

The online version of this article includes the following figure supplement(s) for figure 1:

**Figure supplement 1.** T helper (TH) TH1, TH2, and TH17 peripheral blood phenotyping.

history of developmental difficulties, dental, skin or any musculoskeletal abnormalities. He was an only child from non-consanguineous parents with no family history of PID. Shortly after presenting to immunology, he developed a left hip and iliac fossa pain and was found to have multiloculated left psoas abscess which required a computerized tomography (CT) guided drainage and prolonged antibiotic treatment. This infective episode was complicated by pulmonary embolism (incidental finding on a repeat CT scan) for which he was anticoagulated for a period of 6 months. He had no subsequent history of thromboembolic events. A detailed CT scan of his chest showed cylindrical bronchiectasis of the left lower lobe, lingua and R middle lobe. This appearance remains unchanged as shown on a repeat scan 11 years later in 2016 (*Figure 1A*).

The initial immunological investigations showed profound panhypogammaglobulinaemia with drastically reduced IgG and undetectable IgA and IgM levels (*Table 1*). The basic lymphocyte profiling showed mild CD4⁺ T cell lymphopenia with adequate numbers of B and NK cells. Subsequent detailed phenotyping revealed reduced proportion of naïve T cells and seemingly adequate percentages of central and effector memory T cells. The proportion of TH1/TH2 and TH17 T cells was comparable to healthy controls (*Figure 1—figure supplement 1*) and TCR phenotyping showed a normal TCR repertoire (*Figure 1B*). The patient also had reduced proportion of class switched memory B cells (*Table 1*). Functional testing showed reduced T cell proliferation to both PHA and anti-CD3 stimulation (*Figure 1C and D*). Over the next 15 years, there was evidence of progressive CD4⁺ T cell and B cell lymphopenia. Interestingly, biopsy of his large bowel, which was performed in 2011 for investigation of intermittent diarrhoea, showed mild non-specific inflammatory changes, but also absence of the plasma cells.

Following the initial assessment, the patent was given a provisional diagnosis of common variable immunodeficiency (CVID) and treated with regular intravenous immunoglobulin (IVIG) replacement therapy. His clinical progress since the diagnosis has been uncomplicated. His chronic diarrhoea resolved following commencement of IVIG and the ongoing treatment with IVIG has resulted in excellent control of respiratory infections.

## A rare biallelic variants in *CRACR2A* identified by exome sequencing

The patient was consented for genetic testing and whole-exome sequencing was performed. Full details regarding candidate variant selection are provided in methods. Three potentially damaging rare variants were identified in CRACR2A GRCh37 12:3715799–3873985 ENSG00000130038 ENST00000440314 (c.430A > G; p.R144G, c.898G > T; p.E300*, exons 6/20 and 10/20), which were maternal and c.834G > T; p.E278D in exon 9/20 on the other allele. Only c.834G > T is present in the gnomAD database, though it is considered rare (heterozygous only; 0.00095 in East Asian). All three amino acids are highly conserved in other species.

The variants were confirmed in the genomic DNA by Sanger sequencing the proband and the mother and maternal grandmother and the father, with the pedigree shown in *Figure 2A*. Compound heterozygous segregation of these variants was confirmed by subcloning of the proband cDNA to isolate the individual alleles. Sanger sequencing confirmed the c.834 G > T variant on a separate allele to the other variants (*Figure 2B*). The protein domains affected by the mutations are schematically depicted in *Figure 2C*.

To check whether the mutant proteins are expressed, we performed immunoblot analyses using T cells from patient and healthy control. Peripheral blood mononuclear cells (PBMCs) harvested from the patient and healthy control were stimulated with plate-coated anti-CD3 and anti-CD28 antibodies for 48 hr, following which they were further expanded for a further 4 days and then harvested for making cellular lysates. Western blot analysis using anti-CRACR2A antibodies detected reduced amounts of only the higher molecular weight band (corresponding to the long isoform) in patient samples (*Figure 2D*, left panel). The antibody was targeting the N-terminal 160 amino acids of CRACR2A, and hence it should detect the ~30 KDa truncated form of the p.E300* variant, if expressed. However, we could not detect presence of any band at ~30 KDa, suggesting possibly degradation of the truncated form. To check for CRACR2A transcript levels, we used primers that were common to both isoforms, however, we detected pronounced reduction in CRACR2A transcript levels from patient samples (*Figure 2D*, right panel). Further, in healthy control samples, expression of CRACR2A transcripts increased after stimulation with PMA and ionomycin by more than two-fold, but it was not upregulated in patient T cells. These data suggest possible nonsense-mediated mRNA decay for CRACR2A transcripts.

**Table 1.** Immunological characteristics of patient with variants in EFCAB4B.

| Date/year | | 2005 | 2011 | 2017 | 2019 | 2021 (Feb) | 2021 (Nov) | Ref |
|---|---|---|---|---|---|---|---|---|
| Total lymphocyte count | | 2,880 | 990 | 1,201 | 1,572 | 1,472 | 1,268 | 1000–2800 cells/µl |
| CD3+ T cells | | 2,505 | 733 | 891 | 990 | 952 | 846 | 700–1200 cells/µl |
| CD4+ T cell | Total | 216 | 235 | 148 | 165 | 122 | 112 | 300–1400 cells/µl |
| | CD27+ CD45RA+ (naïve)% | | | 20.92 | 12.65 | 25.5 | | 14–65% |
| | CD27+ CD45RA- (memory)% | | | 42.34 | 47.04 | 42.9 | | 15–52% |
| | CD27-CD45RA- (memory effector) % | | | 36.01 | 37.67 | 20.5 | | 8–35% |
| | CD27+ CD45RA- | | | | 2.64 | 11.1 | | 0–22% |
| | CD25+ CD127- (Treg)% | | | 0.73 | 1.6 | 2.9 | 3.5 | 3–10% |
| CD8+ T cells | Total | 2080 | 475 | 738 | 840 | 772 | 689 | 200–900 cells/µl |
| | CD27+ CD45RA+ (naïve) | | | | 17.69 | 31.3 | | 1.5–65.5% |
| | CD27- CD45RA- (memory) | | | | 25.16 | 19.1 | | 3.5–28.6% |
| | CD27- CD45RA- (memory effector) | | | | 16.09 | 11.1 | | 0.7–72.6% |
| | CD27- CD45RA+ (effector) | | | | 41.07 | 38.6 | | 1.6–53% |
| CD19+ (B cells) | Total | 245 | 102 | 73 | 77 | 87 | 71 | 100–500 cells/µl |
| | CD27- IgM+ IgD + (naïve) % | | | 63.1 | 89.18 | 67.7 | 78 | 44–84% |
| | CD24hi CD38hi (Transitional) % | | | 6.7 | 10.3 | 8.2 | 11 | 2–14% |
| | CD27+ IgM + IgD+ (non-switched) % | | | 15.4 | 3.69 | 22.5 | 11 | 5–32% |
| | CD27+ IgM- IgD- (switched memory) % | | | 14.7 | 7.15 | 5.7 | 5 | 5–33% |
| | Plasmablasts % | | | 0.2 | 0.4 | 1.1 | 1 | 0.2–5% |
| CD56+ CD16+ (NK cells) | | 144 | 139 | 290 | 486 | 429 | 340 | 90–600 cells/µl |
| Immunoglobulin profile | | | | | | | | |
| IgG | | 1.4 | 8.4 | 10.3 | 8.3 | 9.9 | | 6–16 g/l |
| IgA | | <0.06 | <0.06 | <0.06 | <0.06 | <0.06 | | 0.8–4 g/l |
| IgM | | <0.05 | <0.05 | <0.05 | <0.05 | <0.05 | | 0.5–2 g/l |
| IgE | | | | | <0.2 | | | 1–120 ku/l |

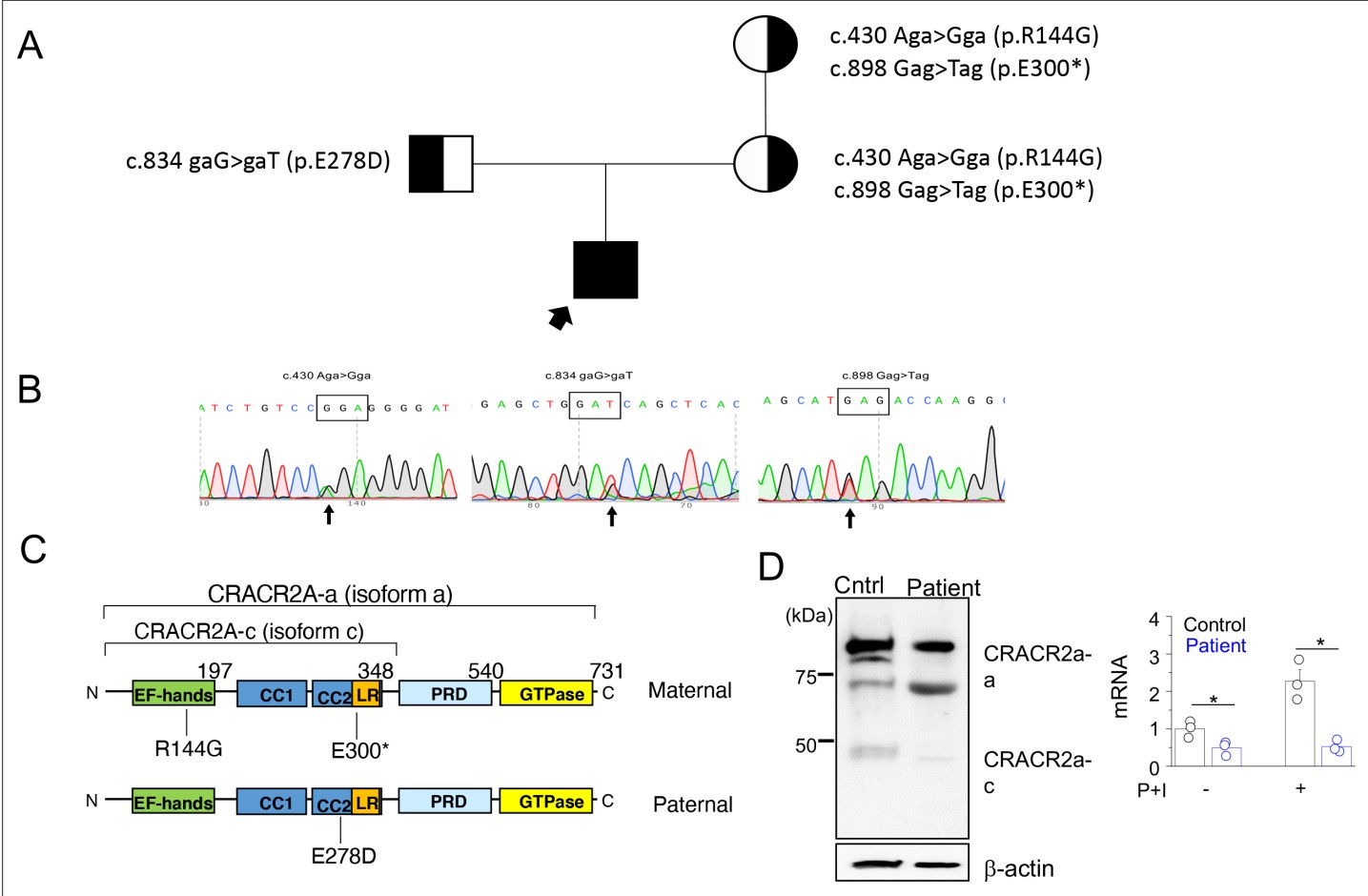

**Figure 2.** A rare biallelic mutation in CRACR2A segregates with the disease. (**A**) Pedigree showing the index case and p.[R144G/E300*] and p.E278D CRACR2A variant carriers. The circles and squares represent female and male members, respectively. Black shading shows affected individuals, either heterozygous (partial black) or homozygous (completely black). (**B**) Sanger sequencing data of the proband showing the indicated mutations. (**C**) Schematic of human CRACR2A showing the location of compound heterozygous mutations from the patient's alleles. The maternal allele comprised of an p.R144G variant that was located in the EF-hand motif and an p.E300* truncation variant within the leucine-rich region (LR). The paternal allele contained the p.E278D variant that was located in the coiled-coil domain 2 (CC1 and CC2). CRACR2A also contains a proline-rich domain (PRD) and a C-terminal Rab GTPase domain. (**D**) Detection of CRACR2A in human PBMCs harvested from healthy control and patient by immunoblotting (left) and real-time quantitative RT-PCR (right). For qRT-PCR, cells were left unstimulated or re-stimulated with PMA plus ionomycin for 5 hr. β-actin was used as a loading control for immunoblotting. Immunoblot is representative of two independent experiments and mRNA analysis shows representative triplicate from two independent experiments. * $p < 0.05$.

## Patient B cells exhibit intact plasma cell generation

Since the patient exhibited profound panhypogammaglobulinaemia and a relative decline in memory B and plasma cell populations, we next checked whether patient B cells were functional. To test if the panhypogammaglobulinaemia is due to an intrinsic B cell defect, we set up an in vitro culture system with a T cell-dependent (TD) stimulus that enables the generation of mature plasma cells from primary B cells (*Figure 3A*; *Cocco et al., 2012*). The capacity of the patient's B cells to differentiate into plasma cells was comparable to healthy controls, as judged by the acquisition of CD38 and CD138 surface expression, which is associated with the transition to an antibody-secreting state (*Figure 3B*). Furthermore, assessment of immunoglobin levels in cell culture showed adequate production of IgM and IgG (*Figure 3C*), suggesting that the failure of antibody production in this case might be secondary to inadequate in vivo T cell help.

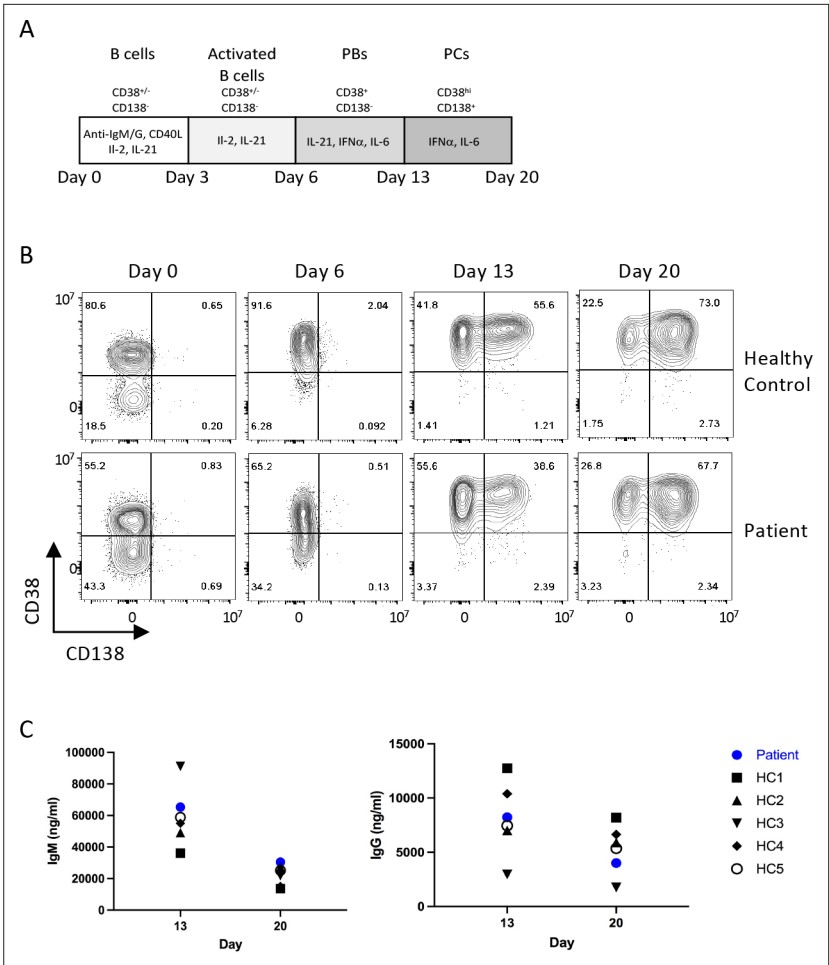

**Figure 3.** Patient B cell responses to T cell-dependent (TD) stimulation. (**A**) Schematic representation of the culture conditions used to assess plasma cell differentiation in vitro. PBs – Plasmablasts, PCs – Plasma cells (**B**) Representative flow plots showing CD38 and CD138 expression profile following TD stimulation with F(ab') two anti-IgG/M and CD40L. Percentages are indicated within individual quadrants. Data are representative of 2 independent assays. (**C**) Secreted levels of IgM (left) and IgG (right) from the cells in (**B**) at indicated time points. HC1-5 (Healthy control) represents data from five independent healthy donor cells.

## Patient T cells show reduced cytokine expression due to decreased Ca²⁺ influx and JNK activation

Since the patient B cells produced antibodies in vitro, in response to TD antigen, we surmised that panhypogammaglobulinaemia is likely due to defect in T cell function. Previously we showed that *Cracr2a* KO mice had a defect in effector of T cell functions (*Woo et al., 2018*). Hence, we further analyzed phenotypes of patient T cells, including TCR-proximal signaling events and cytokine production. Patient CD4⁺ cells produced less IFN-γ than the ones isolated from a representative healthy donor (*Figure 4A*). We extended our findings using cells from multiple healthy donors and different batches of the patient cells by checking expression of IFN-γ, IL-2, and TNF using quantitative RT-PCR (*Figure 4B*) and ELISA (*Figure 4C*). Expression of all these cytokines was impaired in patient T cells compared to control T cells derived from multiple different healthy donors. Since CRACR2A is known to interact with ORAI1 and STIM1, to regulate SOCE and JNK phosphorylation (*Srikanth et al., 2010a*; *Srikanth et al., 2016*), we measured SOCE after anti-CD3 cross-linking as well as passive depletion of the ER Ca²⁺ stores using thapsigargin, a SERCA blocker. In both these analyses, patient cells showed significant reduction in SOCE, when compared to those from representative healthy controls (*Figure 4D and E*). Further, patient T cells also showed a profound defect in JNK phosphorylation upon TCR cross-linking (*Figure 4F*). To confirm that reduced SOCE in patient cells is not due to

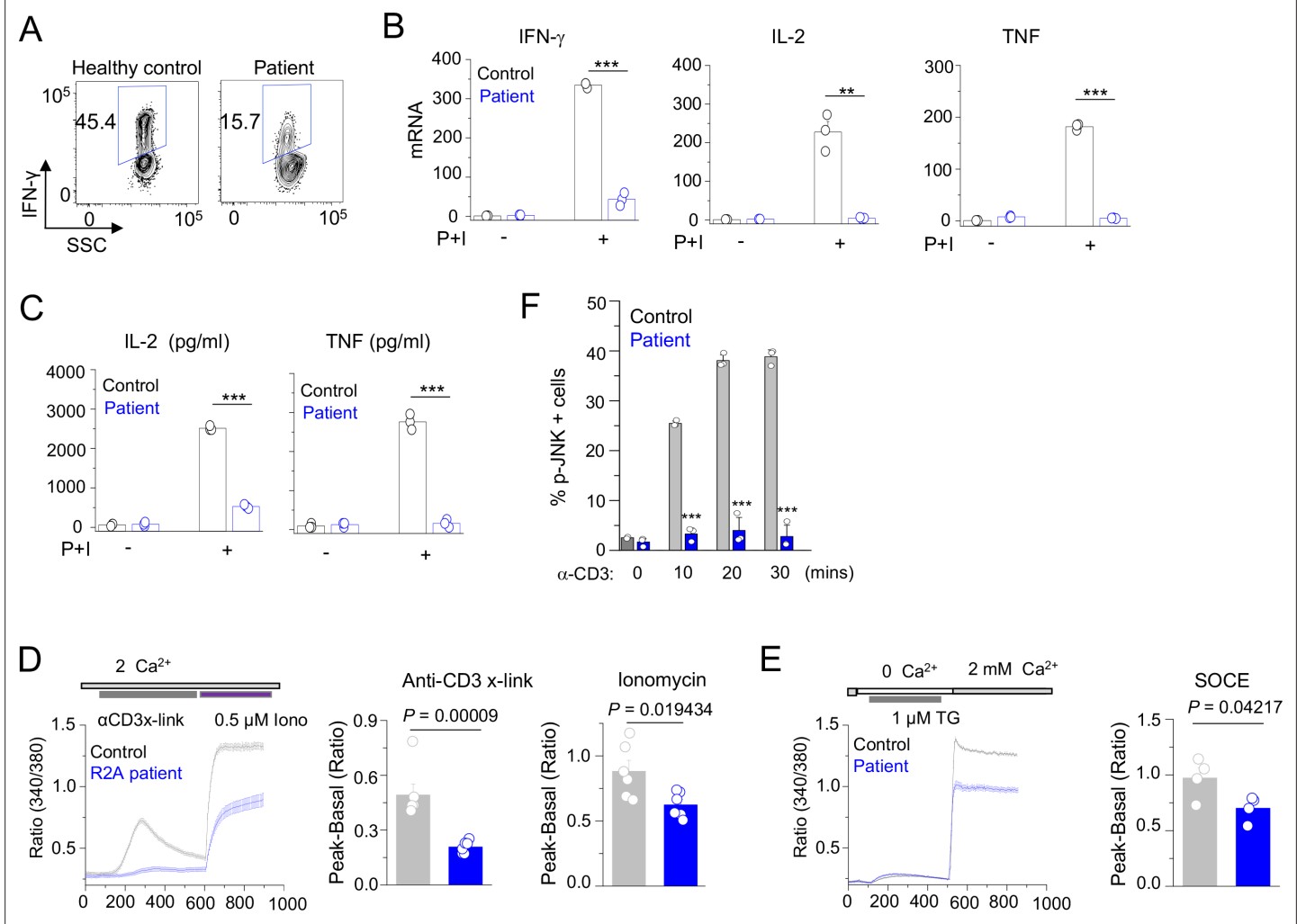

**Figure 4.** Patient T cell responses to T cell receptor stimulation. (**A**) Representative flow plots showing expression of IFN-γ in human PBMCs from a healthy donor and the patient. PBMCs were stimulated with anti-CD3 and anti-CD28 antibodies for 48 hr and cultured for further 4 days in the presence of IL-2 before re-stimulation with PMA plus ionomycin for 5 hr for cytokine analysis. Cells were gated for CD4+ T cells. (**B**) Quantitative mRNA expression analysis (± s.d.m.) of indicated cytokines from human PBMCs (cultured as mentioned above) with or without stimulation with PMA plus ionomycin for 5 hr. (**C**) Levels of IL-2 and TNF in human PBMCs from culture supernatants of cells stimulated as described above (**B**) were determined by ELISA. (**D**) Representative traces showing averaged (± SEM) SOCE responses from healthy control and patient PBMCs (cultured as indicated in A), after transient stimulation with anti-CD3 antibody cross-linking, or ionomycin (0.5 μM) in the presence of external solution containing 2 mM $Ca^{2+}$ (left) as indicated. Bar graphs show baseline subtracted ratio values for anti-CD3 antibody cross-linking or ionomycin (average± SEM) from six independent experiments (right). (**E**) Representative traces showing averaged (± SEM) SOCE responses from healthy control and patient PBMCs (cultured as indicated in A), after store-depletion with thapsigargin (1 μM) stimulation in $Ca^{2+}$-free Ringer's solution. SOCE was measured by addition of 2 mM $Ca^{2+}$-containing Ringer's solution as indicated (left). Bar graph shows baseline subtracted ratio values at the peak of SOCE (average± SEM) from four independent experiments (right). (**F**) Phosphorylated JNK levels in CD4+ T cells from healthy control and patient PBMCs (cultured as indicated in A), stimulated with anti-CD3 antibody for indicated times. Bar graphs show average± SEM from three independent experiments. ** p < 0.005, *** p < 0.0001.

The online version of this article includes the following figure supplement(s) for figure 4:

**Figure supplement 1.** Expression of ORAI1 and STIM1 in PBMCs from healthy control and patient.

altered expression of CRAC channel subunits, we checked expression of ORAI1 and STIM1 in lysates of T cells from healthy controls and patient. Expression of ORAI1 and STIM1 were similar between control and patient T cells as judged by immunoblotting as well as real-time quantitative RT-PCR measurements (***Figure 4—figure supplement 1***). Together, these results show the conserved role of CRACR2A in effector T cell responses by regulating the $Ca^{2+}$ and JNK pathways in human effector T cells, consistent with the previous finding in murine T cells (***Woo et al., 2018***).

## Mutations of E278D and R144G/E300* lead to reduced cytokine production in T cells

To gain insight into the phenotypes of mutants derived from individual *CRACR2A* alleles in the patient, we first cloned the mutant cDNA in a lentiviral vector. When overexpressed in HEK293 cells, E278D and R144G/E300* mutants migrated at the expected molecular weights of ~90 and ~ 30 kDa, respectively (*Figure 5—figure supplement 1A*). To check their expression without interference from endogenous CRACR2A proteins, we generated *CRACR2A* KO Jurkat T cells using the lentivirus-based CRISPR/Cas9 system and found that sgRNA#1 and #2 had a great deletion efficiency (*Figure 5— figure supplement 1B*). When we stably expressed E278D and R144G/E300* mutants in *CRACR2A* KO Jurkat cells (sgRNA#2), these mutants migrated at their expected molecular weights, similarly

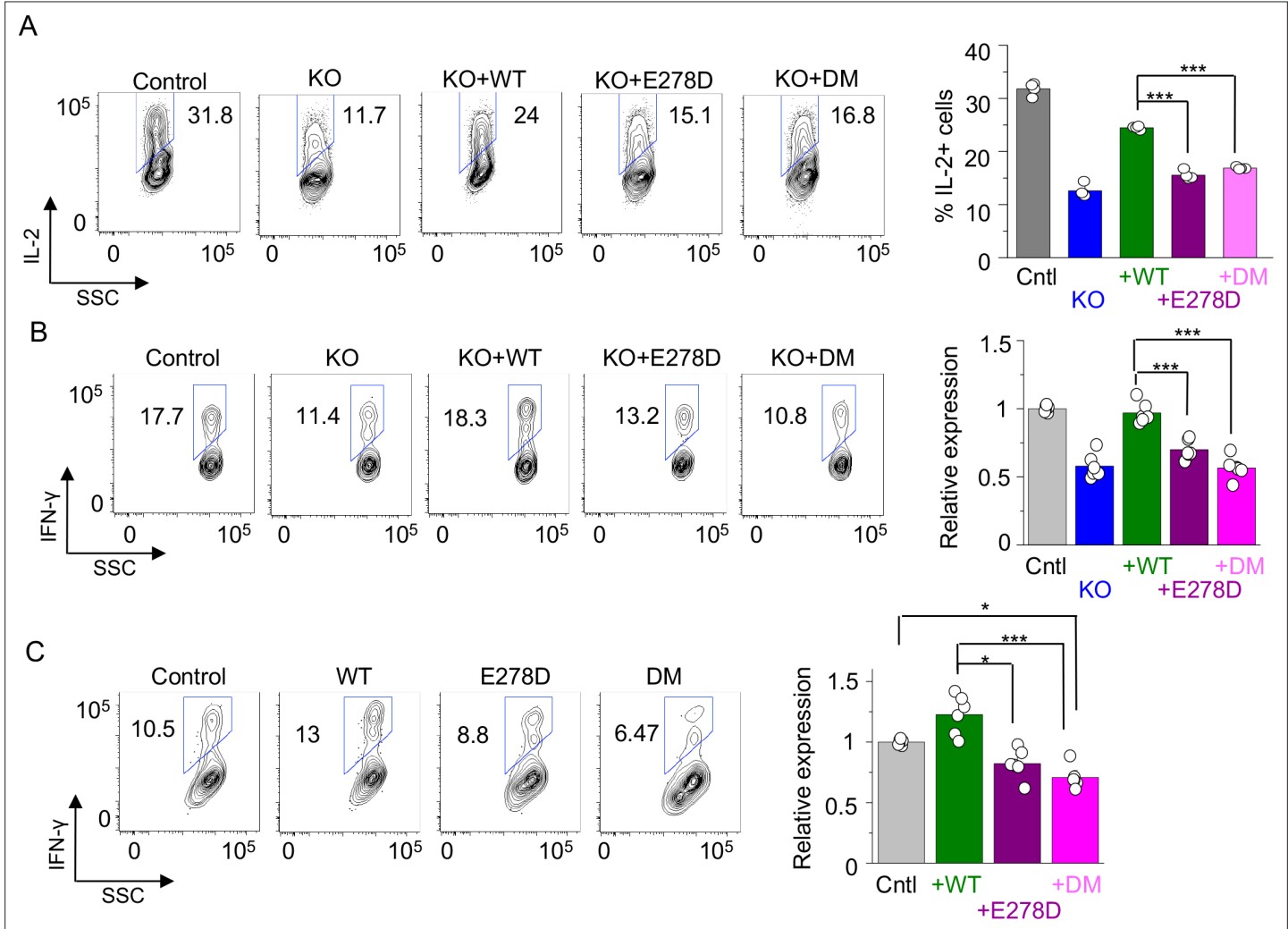

**Figure 5.** Cytokine expression profiles in T cells expressing individual allelic mutations in CRACR2A. (**A**) Representative flow plots showing expression of IL-2 in control or *CRACR2A* KO Jurkat T cells stably expressing FLAG-tagged WT CRACR2A, CRACR2A$^{E278D}$ (E278D), or CRACR2A$^{R144G/E300*}$ (DM) after stimulation with PMA plus ionomycin for 16 hr (left). Bar graph shows means ± s.e.m. of pooled technical replicates from two independent experiments (right). (**B**) Representative flow plots showing expression of IFN-γ in primary human CD4+ T cells transduced with lentiviruses encoding *CRACR2A*-targeting sgRNA and those encoding cDNAs of WT or indicated mutants of CRACR2A after stimulation with anti-CD3 and anti-CD28 antibodies for 5 hr (left). Bar graph (right) shows means ± s.e.m. of pooled technical replicates from three independent experiments. (**C**) Representative flow plots showing expression of IFN-γ in primary human CD4+ T cells purified from healthy donors and transduced with lentiviruses encoding cDNAs for WT or indicated mutant of CRACR2A after stimulation with anti-CD3 and anti-CD28 antibodies for 5 hr (left). Bar graph (right) shows means ± s.e.m. of pooled technical replicates from three independent experiments. * p < 0.05, *** p < 0.0001.

The online version of this article includes the following figure supplement(s) for figure 5:

**Figure supplement 1.** Expression profile of individual allelic mutants of CRACR2A in different cell types.

with HEK293 cells. Notably, even after repeated transduction the expression of R144G/E300* mutant protein was much lower than WT or E278D CRACR2A, suggesting that the transcripts of this double mutant may be unstable, similar to our observation with patient cells (*Figure 2D*).

Similar to patient T cells, CRACR2A KO Jurkat T cells showed reduced IL-2 production (*Figure 5A*). Stable expression of WT CRACR2A in these cells substantially rescued the IL-2 production defect, validating that the decrease was caused due to loss of CRACR2A. However, expression of E278D or R144G/E300* mutants showed very marginal rescue of IL-2 expression in CRACR2A KO Jurkat T cells under the same condition. We also validated these results with primary human CD4$^+$ cells. We found that transduction with lentiviral vectors encoding sgRNA to delete *CRACR2A* resulted in reduced IFN-γ production in primary cells, which was completely rescued by expression of WT CRACR2A (*Figure 5B*). However, both E278D and R144G/E300* mutants did not rescue cytokine production in the KO primary T cells, similar to our observations with Jurkat T cells. Exogenous expression of WT CRACR2A in primary T cells from healthy donors (in the presence of WT CRACR2A) slightly enhanced IFN-γ production while the mutants did not (*Figure 5C*). On the contrary, overexpression of the R144G/E300* showed significant reduction in cytokine production, suggesting that this mutant may impair function of endogenous CRACR2A. Overall, our cytokine analysis suggested that expression of mutants derived from each of the *CRACR2A* alleles could not rescue cytokine production in CRACR2A KO T cells.

## Mutations of E278D and R144G/E300* lead to reduced Ca²⁺ influx and JNK activation

We next checked whether individual mutants could rescue TCR-proximal events in CRACR2A KO T cells. Earlier we showed that loss of CRACR2A impaired TCR stimulation-induced SOCE and JNK phosphorylation in T cells (*Srikanth et al., 2016*). Accordingly, CRACR2A KO Jurkat T cells showed reduced Ca²⁺ entry triggered by TCR stimulation (*Figure 6A*). Expression of WT CRACR2A fully rescued the decreased level of Ca²⁺ entry in KO cells while that of R144G/E300* and E278D mutants did not. We observed similar results when SOCE was triggered by passive depletion of intracellular Ca²⁺ stores using thapsigargin (*Figure 6B*). Interestingly, these two mutants behaved differently in activation of the JNK signaling pathway. While WT and the E278D mutant rescued phosphorylation of JNK, R144G/E300* mutant failed to do so (*Figure 6C*). These results indicate that while R144G/E300* mutant has a defect in both Ca²⁺ entry and JNK activation due to truncation of the coiled coil, proline-rich and Rab GTPase domains, E278D mutant has a selective defect in Ca²⁺ signaling.

## CRACR2A$^{R144G/E300*}$ has defect in the interaction with downstream regulators and shows abnormal cytoplasmic distribution

The C-terminal Rab GTPase domain of CRACR2a contains the conserved residues required for GTP binding and hydrolysis as well as a prenylation site for membrane anchoring (*Figure 2C*). The N-terminal EF-hand motif and coiled-coil domains are involved in sensing intracellular Ca²⁺ levels and stabilizing ORAI1-STIM1 interaction (*Srikanth et al., 2010a*). Also, the longer isoform, CRACR2A, which localizes to intracellular vesicles, translocates to the immunological synapse, and interacts with a signaling adaptor molecule, Vav1, through its proline-rich domain (PRD) (*Srikanth et al., 2016*).

To gain insight into the mechanism underlying the defective function of the mutants, we checked the interaction of these mutants with three known interacting partners, ORAI1, STIM1, and Vav1 using co-immunoprecipitation. These results showed that the R144G/E300* mutant failed to interact with all the molecules, whereas the E278D mutant retained binding to all three partners, similar to WT CRACR2A (*Figure 7A and B*). Confocal analysis showed that WT CRACR2A and E278D mutant localized in the proximal Golgi area and translocated to the interface of anti-CD3 antibody-coated coverslips after stimulation (*Figure 7C*). However, the R144G/E300* mutant showed a diffuse cytoplasmic localization and was not enriched at the interface in stimulated cells. These results suggest that while E278 behaves similar to WT CRACR2A in interacting with binding partners (under our experimental conditions) as well as intracellular localization under resting and stimulated conditions, the R144G/E300* mutant is impaired in all these functions. Deletion of protein interaction domains including the PRD and the C-terminal prenylation site is likely the cause loss of binding to its partners and cytoplasmic localization of the DM. Even though E278D retained binding to ORAI1/STIM1, it could not reconstitute SOCE or cytokine production in *CRACR2A* KO Jurkat T cells, suggesting that subtle

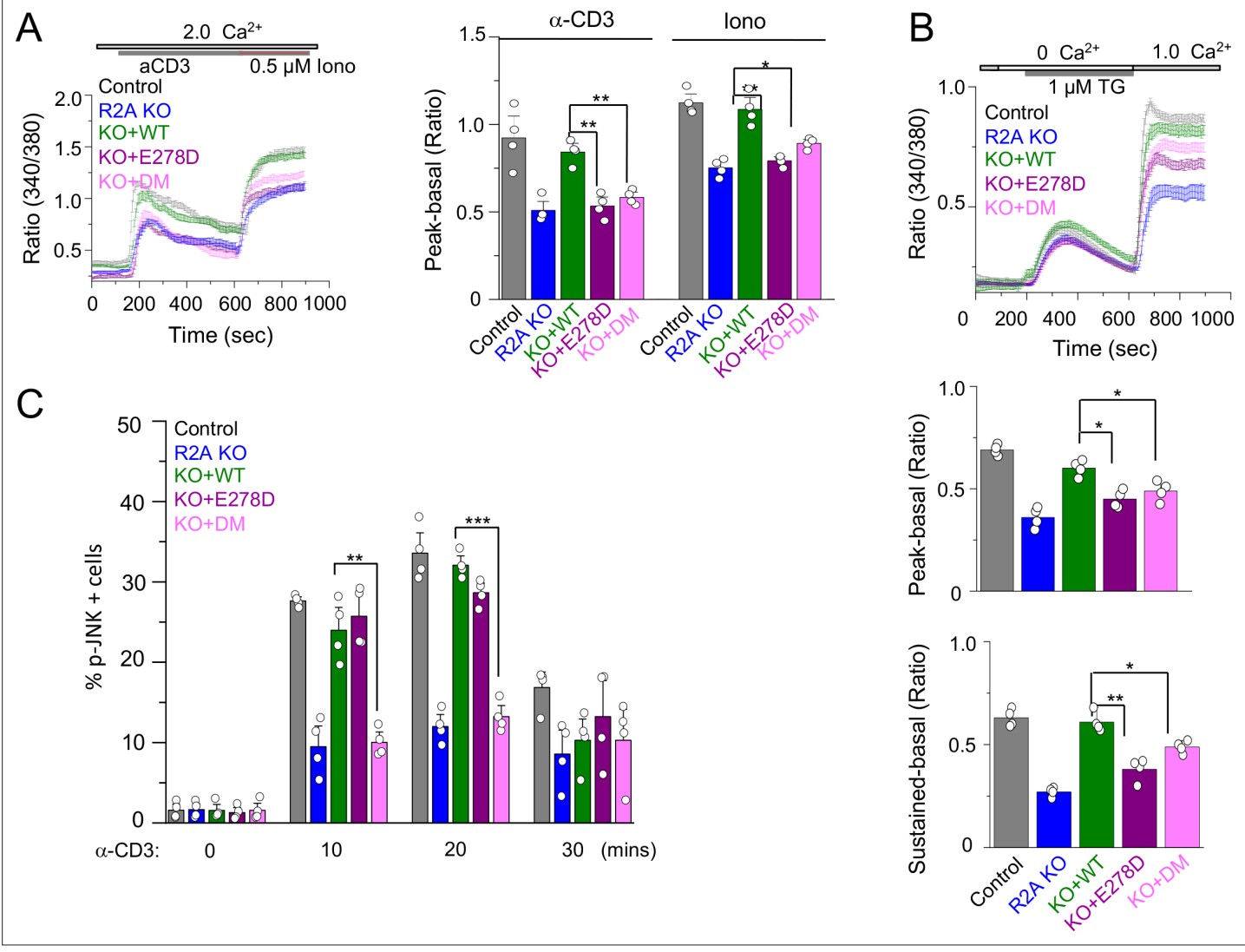

**Figure 6.** SOCE and JNK phosphorylation in T cells expressing individual allelic mutations in CRACR2A. (**A**) Representative traces showing averaged SOCE from control (48 cells) or CRACR2A-KO Jurkat T cells (KO) transduced with an empty vector (55 cells) or those encoding FLAG-tagged WT CRACR2A (45 cells), CRACR2A$^{E278D}$ (E278D, 40 cells), or CRACR2A$^{R144G/E300*}$ (DM, 50 cells) mutants (left). Cells were stimulated with anti-CD3 antibodies, followed by ionomycin treatment in the presence of external solution containing 2 mM Ca$^{2+}$. Bar graphs show averaged baseline subtracted peak SOCE (± s.e.m.) from anti-CD3 antibody and ionomycin treatments, from three independent experiments (right). (**B**) Representative traces showing averaged SOCE induced by thapsigargin (TG) treatment from control (52 cells) or CRACR2A-KO Jurkat T cells (KO) transduced with empty vector (46 cells) or those encoding FLAG-tagged WT CRACR2A (49 cells), CRACR2A$^{E278D}$ (E278D, 51 cells), or CRACR2A$^{R144G/E300*}$ (DM, 50 cells) mutants (top). Cells were stimulated with thapsigargin in Ca$^{2+}$-free solution to deplete the intracellular stores and exposed to external solution containing 2.0 mM Ca$^{2+}$. Bar graphs below show averaged baseline subtracted SOCE levels (± s.e.m.) at the peak (center) or later time point (sustained – 900 s, bottom) from three independent experiments. (**C**) Phosphorylated JNK levels in control or CRACR2A KO Jurkat T cells stably expressing WT and indicated mutants of CRACR2A, stimulated with anti-CD3 antibody for indicated times. Graphs show average± SDM from three independent experiments. * p < 0.05, ** p < 0.005, *** p < 0.0001.

changes in the protein conformation due to this mutation may alter its affinity for interaction with ORAI1/STIM1 under physiological conditions, which is not recapitulated in our biochemical studies performed using overexpression.

## Discussion

Our study describes a novel cause of primary immunodeficiency in humans due to compound hetero-zygous mutations in *CRACR2A*. The resulting clinical phenotype is more in keeping with antibody

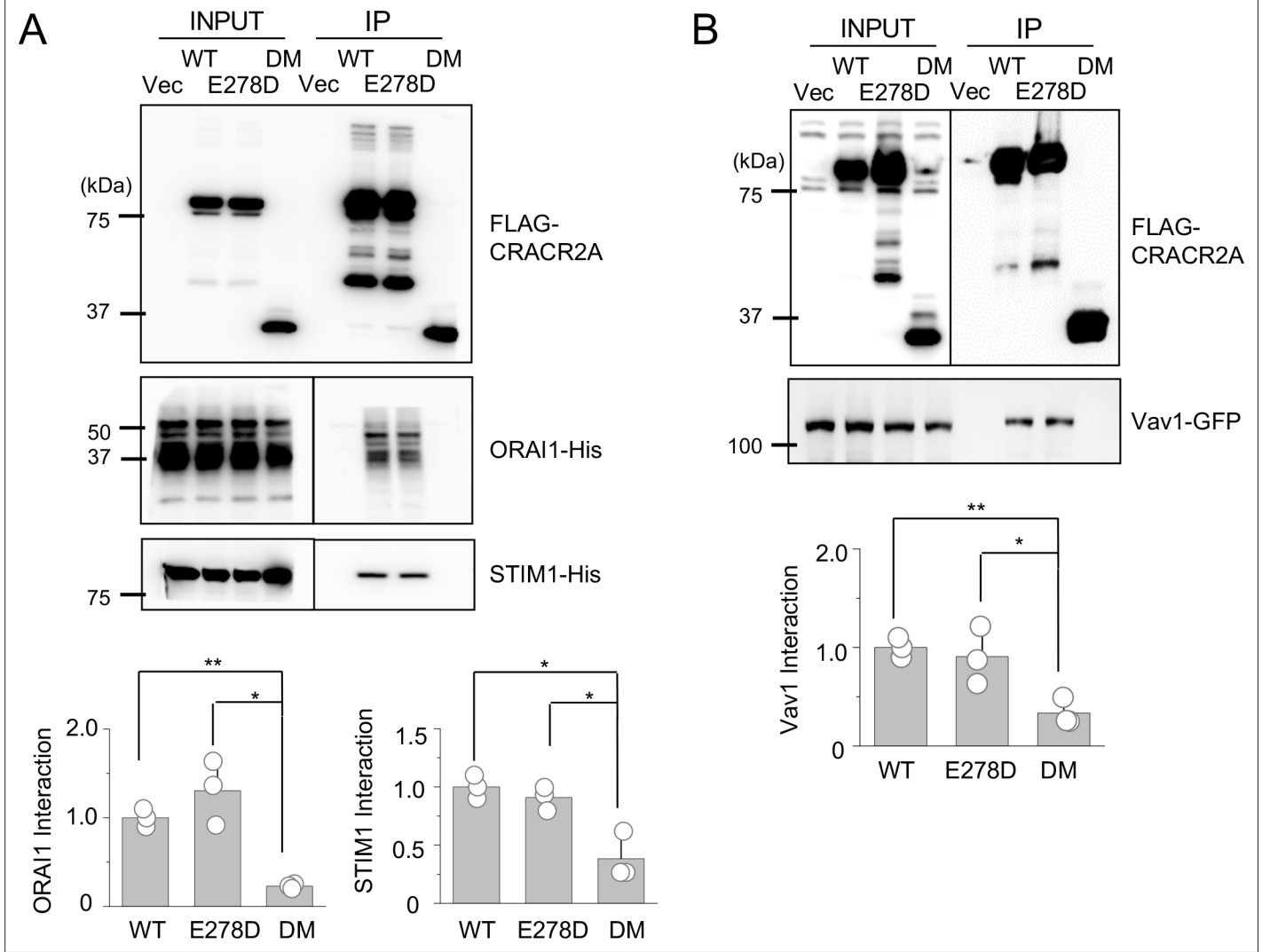

**Figure 7.** Biochemical interactions and localization profiles of individual allelic mutations in CRACR2A. (**A**) Immunoprecipitation for detection of binding between WT and indicated mutants of CRACR2A (E278D, or DM) with ORAI1/STIM1. Lysates of HEK293T cells expressing FLAG-tagged CRACR2A in the presence of 6 X His-tagged ORAI1 and STIM1, were subjected to immunoprecipitation with anti-FLAG antibodies and analyzed by immunoblotting for detection of the indicated proteins. Bar graphs below show densitometry analysis of binding of CRACR2A to ORAI1 (left) and STIM1 (right), normalized to that of WT CRACR2A, from three independent experiments. (**B**) Immunoprecipitation for detection of binding between CRACR2A and VAV1. Lysates of HEK293T cells expressing FLAG-tagged WT or indicated mutants of CRACR2A in the presence of GFP-tagged VAV1, were subjected to immunoprecipitation with anti-FLAG antibodies and analyzed by immunoblotting for detection of indicated proteins. Bar graph below shows densitometry analysis of binding of CRACR2A to VAV1, normalized to that of WT CRACR2A, from three independent experiments. (**C**) Representative confocal images of *CRACR2A* KO Jurkat cells stably expressing N-terminally FLAG-tagged WT or indicated mutants of CRACR2A under resting conditions (top panels) or 20 min after dropping on stimulatory anti-CD3 antibody-coated coverslips (middle and bottom panels). The top panels showed images from the center of the cell. The middle panels show images from the bottom of the cell, which was in contact with the coverslip and the bottom panels showed images from the center of the anti-CD3 antibody-stimulated cell. F-Actin staining – green, anti-FLAG antibody staining – red. Images are representative of at least 10 cells in each condition. * p < 0.05, ** p < 0.005.

deficiency, although detailed immunological investigations show a combined immune defect with evidence of progressive lymphopenia and impaired T cell activation. At presentation, the patient was found to have profound panhypogammaglobulinaemia, however, subsequent investigations suggested that the intrinsic B cell function seems to be relatively preserved in this condition, and that failure of antibody production is most likely a consequence of inadequate T cell help. The same mechanism is likely to cause a reduction in the proportion of class switched memory B cells. Consequently, our investigations have focused on the function of T cells, and in particular CD4+ cells due to the

known role of CRACR2A in their activation and also progressive selective CD4[+] lymphopenia which was evident from the patient's phenotype.

Previously we have characterized developmental and immunological phenotypes in mice with T cell-specific deletion of *Cracr2a*. Using 8–12 week-old animals housed in a specific pathogen free facility with limited exposure to environmental insults, we observed normal development and homeostasis of CD4[+] cells in *Cracr2a*[fl/fl] *Cd4*-Cre mice (*Woo et al., 2018*). Similar to patient cells, CRACR2A KO murine T cells showed defect in SOCE as well as JNK-phosphorylation, thereby, reduction in cytokine production, suggesting conserved function of CRACR2A in mouse and human T cells. Further, within the time frame of 8–12 weeks, no lymphopenia was observed in *Cracr2a*[fl/fl] *Cd4*-Cre mice, unlike the patient in this study, who shows progressive loss of CD4[+] T cells. Further studies examining T cell homeostasis in aged mice can uncover whether this phenotype is conserved in mice. It should be noted that the mice had T cell-specific deletion of *Cracr2a*, unlike the patient, that expresses, albeit at lower amounts, a mutant CRACR2A protein globally. Hence, patient T cells may also have indirect impact on survival, due to loss of CRACR2A in other cells/tissues.

To check the extent of function lost due to individual allelic mutants of CRACR2A, we reconstituted CRACR2A KO T cells with individual mutants. The more severe, truncated R144G/E300* mutant showed a profound impairment in both SOCE as well as JNK phosphorylation, and showed phenotypes similar to CRACR2A KO T cells. Biochemical analysis showed that the R144G/E300* mutant lost binding to all known interacting partners, including ORAI1, STIM1, and Vav1 and had localization pattern different from that of WT CRACR2A, which may contribute to its defect in function. The drastic defect of R144G/E300* mutant was expected because it does not have the C-terminal proline-rich and Rab GTPase domains, which are necessary for protein interaction and subcellular localization of CRACR2A proteins. Consequently, cells expressing the R144G/E300* mutant did not show any rescue of SOCE, JNK phosphorylation or cytokine production. The E178D point mutation on the other hand, could reconstitute the defect in JNK phosphorylation observed in CRACR2A KO T cells, retained binding to ORAI1, STIM1 and Vav1, and showed localization pattern similar to WT CRACR2A. However, in spite of being able to interact with ORAI1 and STIM1 under overexpression conditions, the E278D mutant could not rescue the SOCE defect observed in CRACR2A KO T cells. These data suggest that under physiological conditions with limiting amounts of ORAI1 and STIM1, E278D mutant may have impairment in binding to these proteins, which is not recapitulated in our biochemical experiments using overexpression. Consequently, the E278D mutant also could not rescue the defect in cytokine production observed in CRACR2A KO T cells.

Examination of patient PBMCs showed reduced expression of total CRACR2A protein, corresponding to the long isoform, suggesting that transcription/translation from both alleles contribute to total levels of endogenous CRACR2A protein. Further, in spite of multiple attempts, we could not detect any band corresponding to the R144G/E300* mutant, with an expected MW of ~35 KDa, suggesting that the mutant protein may be unstable and degraded. In support of this hypothesis, exogenous expression analysis suggested that while the E178D mutant was expressed at levels similar to WT, the R144G/E300* double mutant consistently showed lower expression (*Figure 1—figure supplement 1*). Furthermore, *CRACR2A* transcript levels were also significantly reduced at basal level and upon stimulation in patient T cells compared to healthy control. These data suggest that impairment observed in patient T cells can be derived from multiple factors including functional defects and mRNA/protein stabilities.

In this study, we performed extensive characterization of CRACR2A variants and consequently demonstrated significant functional impairment of the patient's T cells. However, the resulting clinical phenotype is possibly less severe than might be expected. This is not necessarily surprising, since discordance between apparent functional immune abnormalities and clinical outcome has also been seen with other inborn errors of immunity caused by mutations in STIM1 (*Rice et al., 2019*). Our study also focused on the effects of CRACR2A deficiency primarily on T cell function. However, our results do not exclude the possibility that CRACR2A can play a role in activation of other innate and adaptive immune cell types, which may have indirect contribution to lymphopenia observed in the patient. Further detailed analysis of other cell types in combination with *Cracr2a* KO mouse models will provide insight into the role of CRACR2A in the immune response. In conclusion, our results suggest that human CRACR2A insufficiency underlies progressive loss of immune competence by uncoupling TCR activation from Ca$^{2+}$ and MAPK signaling. Given the resultant clinical phenotype and

range of associated immunological abnormalities, we propose that CRACR2A deficiency should be included within the group of combined immunodeficiency disorders in the IEI classification.

# Materials and methods

## Key resources table

| Reagent type (species) or resource | Designation | Source or reference | Identifiers | Additional information |
|---|---|---|---|---|
| Gene (*Homo sapiens*) | *CRACR2A* | NCBI | Gene ID: 84,766 | |
| Strain, strain background (*Escherichia coli*) | DH5α | Thermo fisher Scientific | Cat# 18265017 | |
| Cell line (*Homo sapiens*) | Jurkat E6-1 T cells | ATCC | Not tested for mycoplasma | Cat# TIB-152 |
| Cell line (*Homo sapiens*) | HEK293T | ATCC | Not tested for mycoplasma | Cat# CRL-3216 |
| Cell line (*Mus musculus*) | CD40L-expressing L cell fibroblasts | *Diehl et al., 2008* | | Irradiated at 50 Gy for 50 minutes prior to use |
| Cell line (*Mus musculus*) | M2-10B4 bone marrow stromal cells | *Lemoine et al., 1988* | | Irradiated at 57 Gy for 57 minutes prior to use |
| Transfected construct (*Homo sapiens*) | Primers for plasmid construction | This paper | | *Supplementary file 1* |
| Biological sample (*Homo-sapiens*) | Primary human peripheral blood mononuclear cells | This paper | | Isolated from the patient and healthy controls |
| Antibody | Anti-human Phospho JNK (mouse monoclonal) | Cell Signaling Technologies | Cat# 9,255 | Flow cytometry (1:100) |
| Antibody | Anti-Human STIM1(rabbit) | Cell Signaling Technologies | Cat# 5,668 S | WB (1:5000) |
| Antibody | Anti-His-tag (rabbit) | Cell Signaling Technologies | Cat# 12,698 S | WB (1:5000) |
| Antibody | Anti-FLAG tag (mouse monoclonal) | Millipore Sigma | Cat# F3040 | WB (1:5000) |
| Antibody | Anti-human ORAI1 (rabbit polyclonal) | Millipore Sigma | Cat# AB9868 | WB (1:5000) |
| Antibody | Anti-Vav1 (rabbit) | Cell Signaling Technologies | Cat# 2,502 | WB (1:2000) |
| Antibody | Anti-β-actin (mouse monoclonal) | Santa Cruz Biotechnology | Cat# sc-47778 | WB (1:2500) |
| Antibody | anti-human CD3 antibody (mouse monoclonal) | Bio X Cell | Clone OKT-3 | 1 µg/ml |
| Antibody | Anti-human CD28 antibody (mouse monoclonal) | Bio X Cell | Cat. #: BE0921 Clone CD28.2 | 1 µg/ml |
| Antibody | Anti-CD4-FITC (mouse monoclonal) | eBioscience | Clone OKT-4 | FACS 5 µl/test |
| Antibody | Anti-IFN-gama-PE-Cy7 | eBioscience | Clone 45.B3 | FACS 5 µl/test |
| Antibody | Ant-IL2-PE | eBioscience | Cat# MQ1-17H12 | FACS 5 µl/test |

*Continued on next page*

*Continued*

| Reagent type (species) or resource | Designation | Source or reference | Identifiers | Additional information |
|---|---|---|---|---|
| Antibody | Anti-human, CD19-BV421 (mouse monoclonal) | BD | Cat. #: 562,440 Clone HIB19 | FACS 5 µl/test |
| Antibody | Anti-human, CD27-PE (mouse monoclonal) | BD | Cat. #: No:555,441 Clone: M-T271 | FACS 20 µl/test |
| Antibody | Anti-human, IgD-FITC (mouse monoclonal) | BD | Cat. #: 561,490 Clone IA6-2 | FACS 5 µl/test |
| Antibody | Anti-human, IgM-Perc-cy5.5 (mouse monoclonal) | BD | Cat. #: 561,285 Clone G20-R7 | FACS 5 µl/test |
| Antibody | Anti-human, CD24-FITC (mouse monoclonal) | BD | Cat. #: 555,427 Clone ML-5 | FACS 20 µl/test |
| Antibody | Anti-human, CD38-Pecy7 (mouse monoclonal) | BD | Cat. #: 335,825 Clone HB-7 | FACS 5 µl/test |
| Antibody | Anti-human, CD3-V500 (mouse monoclonal) | BD | Cat. #: 561,416 Clone UCHT-1 | FACS 5 µl/test |
| Antibody | Anti-human, CD4-BV421 (mouse monoclonal) | BD | Cat. #: 562,424 Clone RPA-T4 | FACS 5 µl/test |
| Antibody | Anti-human, CD25-Pecy7 monoclonal | BD | Cat. #: 335,824 Clone 2A3 | FACS 5 µl/test |
| Antibody | Anti-human, CD127-PrcP Cy5.5 (mouse monoclonal) | BD | Cat. #: 560,551 Clone HIL-7R-M21 | FACS 5 µl/test |
| Antibody | Anti-human, FoxP3-Alexa 488 (mouse monoclonal) | BD | Cat. #: 566,526 Clone 2632/E7 | FACS 5 µl/test |
| Antibody | Anti-human, CD196-PE (mouse monoclonal) | BD | Cat. #: 559 562 Clone 11A9 | FACS 10 µl/test |
| Antibody | Anti-human, CD-183-APC (mouse monoclonal) | BD | Cat. #: 550,967 | FACS 20 µl/test |
| Antibody | Polyclonal F(ab')two goat anti-human IgM/IgG/IgA | Jackson Immuno Research | Cat. #: 109-006-129 | 10 µg/ml |
| Antibody | Anti-human CD138-APC (mouse monoclonal) | Miltenyi Biotech | Cat. #: 130-117-395 | FACS 2 µl/test |
| Recombinant DNA reagent | FGIIF | This paper | N/A | |

*Continued on next page*

*Continued*

| Reagent type (species) or resource | Designation | Source or reference | Identifiers | Additional information |
|---|---|---|---|---|
| Recombinant DNA reagent | pmCherry-N1 | Clontech | Clontech plasmid #632,523 | |
| Recombinant DNA reagent | pMD2.G | Addgene | Addgene plasmid # 12,259 | |
| Recombinant DNA reagent | psPAX2 | Addgene | Addgene plasmid # 12,260 | |
| Recombinant DNA reagent | pLentiCas9-blasticidin | Addgene | Addgene plasmid # 52,962 | |
| Recombinant DNA reagent | pLentiguide-puro_hCRACR2Asg#1 | This paper | | Details in *Supplementary file 1* |
| Recombinant DNA reagent | pLentiguide-puro_hCRACR2Asg#2 | This paper | | Details in *Supplementary file 1* |
| Recombinant DNA reagent | pLentiguide-puro_hCRACR2Asg#3 | This paper | | Details in *Supplementary file 1* |
| Recombinant DNA reagent | pLentiguide-puro_hORAI1sg | This paper | | Details in *Supplementary file 1* |
| Recombinant DNA reagent | FG11F CRACR2A WT | This paper | | Details in *Supplementary file 1* |
| Recombinant DNA reagent | FG11F CRACR2A$^{E278D}$ | This paper | | Details in *Supplementary file 1* |
| Recombinant DNA reagent | FG11F CRACR2A$^{R144G}$ | This paper | | Details in *Supplementary file 1* |
| Recombinant DNA reagent | FG11F CRACR2A$^{R144G,E300*}$ | This paper | | Details in *Supplementary file 1* |
| Sequence-based reagent | SureSelect XT Human All Exon V5 | Agilent | | |
| Sequence-based reagent | CRACR2A Exon 6_F | This paper | | ATGATTCCT GGCAGGTGAGA |
| Sequence-based reagent | CRACR2A Exon 6_R | This paper | | ATTCCAGTG CAGGGACCAG |
| Sequence-based reagent | CRACR2A Exon 9_F | This paper | | GGCCCTGATG TTGAGTAGGT |
| Sequence-based reagent | CRACR2A Exon 9_R | This paper | | GTGAATGGC AGGGAAAGTGG |
| Sequence-based reagent | CRACR2A Exon 10_F | This paper | | AAACAAGGT GAGGCCAGGG |
| Sequence-based reagent | CRACR2A Exon 10_R | This paper | | AGCCCAAAT CCTCTT TTCACAG |
| Peptide, recombinant protein | IL-2 | Roche | Cat. #: HIL2-RO | |
| Peptide, recombinant protein | IL-21 | Peprotech | Cat. #: 200–21 | 50 ng/ml |
| Peptide, recombinant protein | IL-2 | Peprotech | Cat# 200–02 | 20 units/ml |
| Commercial assay or kit | aCD3 plates | Corning | Cat. #:354,725 | 10 ng/ml |
| Commercial assay or kit | B cell isolation kit | Miltenyi Biotech, | Cat. #:130-091-151 | 100 U/ml |

*Continued on next page*

*Continued*

| Reagent type (species) or resource | Designation | Source or reference | Identifiers | Additional information |
|---|---|---|---|---|
| Commercial assay or kit | QIAamp DNA Blood kit | Qiagen | Cat. #:61,104 | |
| Commercial assay or kit | Human IgG ELISA Quantitation Set | Bethyl Laboratories Inc, | Cat. #: E80-104 | |
| Commercial assay or kit | Human IgM ELISA Quantitation Set | Bethyl Laboratories Inc, | Cat. #: E80-100 | |
| Commercial assay or kit | MagniSort human naïve CD4+ T cell enrichment kit | Thermofisher Scientific | Cat. #: 8804-6814-74 | |
| Commercial assay or kit | FOXP3/ Transcription Factor staining Buffer set | Thermofisher Scientific | Cat. #: 2229155 | |
| Chemical compound, drug | PHA-L | Sigma | Cat. #: L-4144 | |
| Chemical compound, drug | 3HThymidine | Perkin Elmer | Cat. #: 027001 | |
| Chemical compound, drug | LymphoPrep | Axis Shield | Cat. #: 1114547 | |
| Chemical compound, drug | 7-AAD-PerCP-Cy5 | BD | Cat. #: 559,925 | FACS 5 µl/test |
| Chemical compound, drug | Fura 2-AM | Thermofisher Scientific | Cat# F1221 | |
| Chemical compound, drug | Brefeldin A | Thermofisher Scientific | Cat# 00-4506-51 | |
| Chemical compound, drug | Thapsigargin | EMD Millipore | Cat# 80055–474 | |
| Chemical compound, drug | Phorbol 12-myristate 13-acetate (PMA) | EMD Millipore | Cat# 5.00582.0001 | |
| Chemical compound, drug | Ionomycin | EMD Millipore | Cat# 407,951 | |
| Chemical compound, drug | Polybrene | Millipore Sigma | Cat# TR-1003 | |
| Chemical compound, drug | Puromycin | Invivogen | Cat# ant-pr-1 | |
| Chemical compound, drug | Blasticidin | Invivogen | Cat# ant-bl-05 | |
| Chemical compound, drug | Poly-D-Lysine | Thermo Fisher Scientific | Cat# A003E | |
| Chemical compound, drug | Fixable Viability Dye eFluor 780 | eBioscience | Cat# 65-0865-14 | 1 µl/ml of cells |
| Software, algorithm | FlowJo v10 | TreeStar | | |

*Continued on next page*

*Continued*

| Reagent type (species) or resource | Designation | Source or reference | Identifiers | Additional information |
|---|---|---|---|---|
| Software, algorithm | Slidebook software | Intelligent Imaging Innovations, Inc | | |
| Software, algorithm | OriginPro | Originlab | | |
| Software, algorithm | Image J | NIH | | |
| Software, algorithm | Fluoview FV10i Confocal Microscope | Olympus | | |
| Software, algorithm | Fluoview software | FlowJo, LLC | | |
| Software, algorithm | Exome sequence analysis (various) | | | Please see methods section |
| Other | LAS-3000 LCD camera | FujiFilm | | |
| Other | ECM 830 electroporator | BTX | | |
| Other | BD Fortessa flow cytometer | BD Biosciences | | |
| Other | Cytoflex LX flow cytometer | Beckman Coulter | | |
| Other | HiSeq 3,000 | Illumina | | |

Ethical approval for this study was obtained from Leeds East Yorkshire Research ethics Committee (18/YH/0070). Written consent to conduct and publish this study was obtained from all participants.

## Exome sequence preparation

Genomic DNA was purified from whole blood using QIAamp DNA Blood kit (Qiagen) with quality and concentration assayed using a TapeStation (Agilent). Exome sequence data was produced by batch sequencing with other non-related samples. Samples were sonicated using a Covaris E220 Focused-ultrasonicator to fragment the DNA to a median size of approximately 250–300 bp. Libraries were generated using Agilent SureSelect XT Human All Exon V5, using a standard protocol, receiving a unique index sequence and were then pooled to form equimolar pools. These were sequenced on a 150 bp paired end Illumina HiSeq 3,000 (Illumina, San Diego, CA, United States) with an average depth of coverage of x30.

## Candidate variant selection

Exome sequences were trimmed and checked for quality control using CutAdapt (*Andrews, 2010*) and FastQC (*Andrews, 2010*) via Trim-galore (*Krueger, 2015*). Reads were aligned to GrCh37 using BWA-MEM (*Li and Durbin, 2009*). GATKv3 best practices were applied for genomic analysis. Reads were sorted and marked with Picard SortSam and MarkDuplicates (Broad Institute), respectively. The read alignments were readjusted for optimal local alignment using IndelRealigner (GATK), Base Quality Score Recalibration (GATK) was used to estimate base call accuracy by detecting systematic sequencing errors, followed by (GATK) PrintReads. HaplotypeCaller was used to produce gVCFs via local re-assembly of haplotypes. GenotypeGVCFs (GATK) was used for joint genotyping with other non-related samples and in-house controls for an analysis group size of 34 (*Auwera et al., 2018*). Variant Quality Score Recalibration (GATK) was used to filter out probable artifacts from the callset. A total of 719,868 variants were present for the combined cohort, with 49,891 of these variants occurring in the proband sample.

Filtering and prediction of functional consequences was performed by annotating data using Variant Effect Predictor (VEP) (*McLaren et al., 2016*), Exome Variant Server (EVS), The Single Nucleotide Polymorphism database (dbSNP), ClinVar, and the Exome Aggregation Consortium and the Genome Aggregation Database (gnomAD) (*Lek et al., 2016*). Filtering of common variations and annotation terms was performed using the tool VCFhacks (*Parry, 2015*). Candidate variants were required to pass the following filtering conditions: read frequency (count/coverage) between 20 and 100%, according to VEP-annotation at least one canonical transcript is affected with one of the following consequence: variants of the coding sequence, frameshift, missense, protein altering, splice

acceptor, splice donor, or splice region; an in-frame insertion or deletion; a start lost, stop gained, or stop retained, or according to VEP a gnomAD frequency of unknown, less of equal to 0.01, or with clinical significance 'path'.Cohort-specific filtering retained functional variants that were present in at least one case but absent for in-house filtering controls or having indication of misalignment or index swapping. The resulting candidate list of variants consisted of rare variants that were of unknown or potentially damaging significance.

Candidate variants were also annotated with terms from Gene Ontology, GeneRIF, Biogrid inter-action, OMIM, NCBI summary, MGI phenotype, MorbidMap, VOC MammalianPhenotype, UniProtKB, PDB, and HMD human phenotype (*Parry, 2015*). Variants were flagged if they had any annotation related to the disease phenotype, were predicted pathogenic, or were in a gene known to cause immunological responses. A filter for all known PID genes was also used to flag any candidates that may have been otherwise excluded. A total of 48 recessive candidate variants (compound heterozy-gous of homozygous) were identified in 34 genes. Filtering based on gene annotation removed genes with no known immunological significance.

The top gene candidate, calcium release activated channel regulator 2 A (*CRACR2A*) ENSG00000130038 (also annotated by previous gene symbol *EFCAB4B*), contained three variants; ENST00000440314 (1) missense c.430A > G, p.R144G, exon 6/20, novel, 12:3788175, Ref T, Alt C, (2) missense c.834G > T, p.E278D, exon 9/20, gnomAD 0.0001647 rs541326469, 12:3765501 Ref C, Alt A, and (3) stop-gained c.898G > T p.E300*, exon 10/20, novel, 12:3763526 Ref C, Alt A. LDpop Tool via LDlink (with appropriate population EAS) was used to guide interpretation prior to confirmation by genomic DNA sequencing of relatives. Chr12:3763526 (p.E300*) and chr12:3788175 (p.R144G) were not in dbSNP build 151, indicating potential LD, inherited together from one parent and in trans with 12:3765501 (p.E278D). The inheritance pattern (p.[R144G;E300*];[E278D]) was confirmed by Sanger sequencing of genomic DNA in the proband, mother, maternal grandmother, and the father. Ampli-fication of genomic DNA for Sanger sequencing was performed by the standard PCR method. PCR clean-up was performed with ExoSAP-IT (Affymetrix). Sanger sequencing was then done using the same primers; Exon 6, (F- ATGATTCCTGGCAGGTGAGA, R- ATTCCAGTGCAGGGACCAG), Exon 9 (F- GGCCCTGATGTTGAGTAGGT, R- GTGAATGGCAGGGAAAGTGG), and Exon 10 (F- AAACAAG-GTGAGGCCAGGG, R- AGCCCAAATCCTCTTTTCACAG). Sanger sequencing using BigDye Termi-nator Cycle Sequencing Kit, version 3.1 (Applied Biosystems) and analysis on an ABI 3130XL DNA analyzer (Applied Biosystems).

Two candidate variants were also identified in UNC13D, a gene reported as causal for the reces-sive PID Familial hemophagocytic lymphohistiocytosis syndromes (as reported in Genomics England panelapp). However, one variant was present at ~1% in the patient genetic ancestry population and not considered as potentially damaging. Variant 1: ENST00000207549.4 p.Arg411Gln, Polyphen -benign, SIFT – tolerated. East Asian allele frequency 0.008. gnomAD link 1; variant 2: East Asian allele frequency 0. gnomAD link 2 ENST00000207549.4 HGVSpc.753 + 1 G> T, pLoF - high-confidence, East Asian allele frequency 0, gnomAD link 2. No other candidate biallelic variants were considered causal bases on annotation or variant effect. No dominant gain-of-function variants were considered as causal candidates.

## Chemicals and antibodies

Fura 2-AM was purchased from Invitrogen (Carlsbad, CA). Thapsigargin, PMA (Phorbol 12-myristate 13-acetate), and ionomycin were purchased from EMD Millipore. Brefeldin A was purchased from eBioscience. Antibody for detection of CRACR2A (15206–1-AP) was purchased from Proteintech. Antibodies for detection of Vav1 (2502), STIM1 (5668) and His tag (12,698 S) were purchased from Cell Signaling Technologies. Antibodies for detection of FLAG tag (F3040) and ORAI1 (AB9868) were purchased from Millipore Sigma. Antibody for detection of β-actin (sc-47778) was obtained from Santa Cruz Biotechnology.

## Plasmids and cells

Full-length cDNA of human CRACR2A-a (NCBI Reference Sequence: NM_001144958.1) was cloned into a lentiviral vector, FGllF (kind gift from Dr. Dong Sun An, UCLA) with a N-terminal FLAG tag as previously described (*Srikanth et al., 2010a*; *Srikanth et al., 2016*). Various mutants of CRACR2A-a were generated by PCR amplification and site-directed mutagenesis using primers described in

*Supplementary file 1*. All the clones were verified by sequencing. Myc-ORAI1 and Myc-STIM1 plasmids had been described previously (*Gwack et al., 2007*; *Srikanth et al., 2019*). Vav1-GFP clone was purchased from Addgene. HEK293 and Jurkat E6-1 T cell lines were obtained from American Type Culture Collection Center (ATCC, Manassas, VA).

## Human B cell culture, staining, and analysis

PBMCs were isolated from peripheral blood by density dependent centrifugation with Lymphoprep (Axis Shield, Norway). Total B cells were isolated by negative selection with a memory B cell isolation kit according manufacturer's instructions (Miltenyi Biotech, USA). B cells were cultured as previously described (*Cocco et al., 2012*) using gamma-irradiated CD40L-L cells and 2 µg/ml F(ab')two goat anti-human IgM/IgG/IgA (Jackson ImmunoResearch) supplemented with 20 U/ml IL-2 (Roche) and 50 ng/ml IL-21 (Peprotech) as the initial T-depedent stimulus. Cells were monitored for changes in cell surface antigen expression using anti-CD38-APC-Cy7 (HB-7; BD Biosciences), anti-CD138-APC (44F9; Miltenyi Biotec), as well as 7-AAD-PerCP-Cy5 (BD Biosciences) to identify live cells. Flow cytometry was performed using a or a CytoFLEX S or LX (Beckman Coulter). Analysis was performed using FlowJo v10 (TreeStar). Supernatants were collected at day 6 and day 13 from the differentiation assays and assessed for Ig levels using human IgM ELISA Quantitation Set (E80-100) or Human IgG ELISA Quantitation Set (E80-104) (Bethyl Laboratories Inc, USA) according to manufacturer's instructions on a Berthold 96-well plate reader. ELISA absorbance values were analyzed at 450 nm and Ig concentrations calculated from standard curves.

## Generation of CRACR2A-KO Jurkat T cells using CRISPR-Cas9 system

To generate lentiviruses for transduction, HEK293T cells were transfected with plasmid(s) encoding sgRNA and packaging vectors (pMD2.G and psPAX2 – from Addgene) using calcium phosphate transfection method. Cas9 encoding lentivirus was generated using the same technique. Culture supernatants were harvested at 48 and 72 hr post transfection and used for infection (50% of cas9-encoding virus +50% of sgRNA-encoding virus) of Jurkat T cells together with polybrene (8 µg/ml) using the spin-infection method. Cells were selected with puromycin (1 µg/ml) and blasticidin (5 µg/ml) 48 hrs post infection. The sgRNA sequences are described in *Supplementary file 1*.

## Single-cell Ca²⁺ imaging and confocal microscopy

Jurkat or human PBMCs were loaded at $1 \times 10^6$ cells/ml with 1 µM Fura 2-AM for 30 min at 25 °C and attached to poly-D-lysine-coated coverslips. Intracellular $[Ca^{2+}]$ measurements were performed using essentially the same methods as previously described (*Srikanth et al., 2010b*). Confocal laser scanning microscopy was performed using Fluoview FV10i Confocal Microscope (Olympus), images were captured with a 60 x oil objective. Images were processed for enhancement of brightness or contrast using Fluoview software.

## Immunoprecipitation and immunoblotting

For immunoprecipitation, 6 x His-tagged ORAI1 and 6xHis-tagged STIM1 or Vav1-GFP together with empty vector or FLAG-tagged WT CRACR2A-a, FLAG-tagged CRACR2A$^{E278D}$ or CRACR2A$^{R144G, E300*}$ was transfected into HEK293T cells. Transfected cells ($2 \times 10^7$) were lysed in lysis buffer (20 mM Tris-Cl, 2 mM EDTA, 135 mM NaCl, 10% (vol/vol) glycerol, 0.5% Igepal CA-630, protease inhibitor mixture, pH 7.5) and centrifuged at 100,000 x g for 1 hr before preclearing with protein G-Sepharose. Lysates were immunoprecipitated with anti-FLAG antibody-conjugated resin for 6 hr. Immunoprecipitates were washed five times in lysis buffer and analyzed by immunoblotting. For detection of CRACR2A-a, $5 \times 10^6$ HEK293 or CRACR2A-deficient Jurkat T cells stably expressing empty vector, FLAG-tagged WT CRACR2A-a, CRACR2A$^{E278D}$, or CRACR2A$^{R144G, E300*}$ were lysed in in RIPA buffer (10 mM Tris-Cl pH 8.0, 1% Triton X-100, 0.1% SDS, 140 mM NaCl, 1 mM EDTA, 0.1% sodium deoxycholate and protease inhibitor cocktail [Roche]) and centrifuged to remove debris. For immunoblot analyses, lysates were separated on 10% SDS-PAGE and proteins were transferred to nitrocellulose membranes and subsequently analyzed by immunoblotting with relevant antibodies. Chemiluminescence images were acquired using an Image reader LAS-3000 LCD camera (FujiFilm).

## Human T cell culture, staining, and analysis

Peripheral blood mononuclear cells (PBMCs) were obtained under federal and state regulations from the CFAR Virology core Laboratory at UCLA that were prepared from buffy coats from healthy,

unidentified adult donors using Ficoll-PAQUE gradients. PBMCs from healthy control and patient were activated for 48 hr on a plate coated with 10 µg/ml of anti-CD3 antibody (OKT3, Bio X Cell) and cultured in T cell media (DMEM containing 20% fetal bovine serum and 1% Pen-Strep) supplemented with 5 µg/ml of anti-CD28 antibody (Bio X cell), and 20 U/ml IL-2 (Peprotech) for ThN differentiation. The cells were expanded for a further 4 days with IL-2 and on day 6, cells were extensively washed and activated with 20 nM of PMA, 1 µM of ionomycin, and Brefeldin A (3 µg/ml) for 5 hr, surface stained with anti-CD4-FITC, and intracellularly stained with anti-IFN-γ-PE. CD4-FITC-positive cells were gated for analysis. For reconstitution of CRACR2A-a in CRACR2A-a-deficient PBMCs, naive CD4+ T cells were enriched by magnetic sorting from single-cell suspensions using MagniSort naive CD4+ T cell enrichment kit (catalog # 8804-6814-74) according to manufacturer's instructions (ThermoFisher Scientific). Cells were activated and cultured as described above. Cells were infected with lentiviruses encoding cas9 together with those encoding sgRNAs targeting CRACR2A-a on day 1 and with lentiviruses encoding WT CRACR2A-a, CRACR2A$^{E278D}$or CRACR2A$^{R144G, E300*}$ together with empty vector on day 2. 48 hr after infection, cells were selected with 1 µg/ml of puromycin and 5 µg/ml blasticidin for 16 hr and then expanded for further 2 days with fresh media. On day 7, cells were stimulated with 10 µg/ml of anti-CD3 antibody, 5 µg/ml of anti-CD28 antibody and Brefeldin A (3 µg/ml) for 5 hr, intracellularly stained with anti-IFN-γ-PE (4 S.B3). For cytokine staining, 1 × 10$^6$ Jurkat T cells were fixed with 4% p-formaldehyde for 15 mins, permeabilized with 0.5% saponin, blocked and stained with anti-IL-2-PE antibody from Thermofisher Scientific. For flow cytometry, the following human specific antibodies were used: CD4-FITC (OKT4, eBioscience), IFN-γ-PE-Cy7 (45.B3, eBioscience) and IL-2-PE (MQ1-17H12, eBioscience). Data were acquired using FACSCalibur (Becton Dickinson) or BD LSRFortessa cell analyzers and analyzed using FlowJo software (Tree Star). ELISA was performed on human PBMCs harvested from healthy control and patient for detection of IL-2 (ThermoFisher, #88-7025-88) and TNF (ThermoFisher, # 88-7324-88).

## T-(Helper) cell phenotyping

Whole EDTA blood was stained with a combination of CD3-V500, CD4-BV421, CCR6-Pe and CXCR3-Alexa-Fluoro 647 (all antibodies from Becton Dickinson, UK) at room temperature protected from light. Erythrocytes were subsequently lysed using BD red cell lysis solution and following a further 10 min incubation the samples were washed by centrifugation (1500 rmp for 6 min) using PBS/1% FBS. Following the final wash, cells were resuspended in 400 µl of PBS + 0.5% formaldehye and cells analysed using a FACSCanto II flow cytometer (BD) using FACSDIVA software. The gating strategy is show in *Figure 1—figure supplement 1B*.

## RNA isolation, cDNA synthesis, and real-time quantitative PCR

Total RNA from cells harvested in TRIzol Reagent (Thermofisher Scientific) was isolated using the Direct-zol RNA isolation kit (Zymo Research). RNA quantity and quality were confirmed with a NanoDrop ND-1000 spectrophotometer. cDNA was synthesized using 1–2 µg of total RNA using oligo(dT) primers and Maxima Reverse Transcriptase (Thermofisher Scientific). Real-time PCR was performed using iTaq Universal SYBR Green Supermix (Bio-Rad) and an iCycler IQ5 system (Bio-Rad) using gene-specific primers described in *Supplementary file 1*. Threshold cycles (CT) for all the candidate genes were normalized to those of 36B4 to obtain ΔCT. The specificity of primers was examined by melt-curve analysis and agarose gel electrophoresis of PCR products.

## Measurement of JNK phosphorylation by intracellular staining

CRACR2A-deficient Jurkat T cells stably expressing empty vector, FLAG-tagged CRACR2A-a, CRACR2A$^{E278D}$, or CRACR2A$^{R144G, E300*}$ were left untreated or stimulated with 10 µg/ml anti-CD3 antibody (OKT3) for indicated times. Human PBMCs isolated from healthy control and patient were stimulated as described above. Cells were fixed with 4% PFA, permeabilized with ice-cold methanol and stained with phospho-SAPK/Jnk mAb (Cell Signaling, #9257). Cells were washed twice with PBS and analyzed with a FACSCalibur flow cytometer (Becton Dickinson) and FlowJo software.

## Statistical analysis

Statistical analysis was performed using the Origin2018b software (OriginLab, Northampton, MA, USA). Data are presented as mean ± s.e.m. Statistical significance to compare two quantitative

groups was evaluated using two-tailed/unpaired t-test. A critical value for significance of $p < 0.05$ was used throughout the study, and statistical thresholds of 0.05, 0.005 as well as 0.0005 are indicated on the figures by stars (see legends for detail).

## Acknowledgements

The authors thank the patient for agreeing to take part in this study. We also thank the clinical team from the Immunology department at St James's University Hospital Leeds, for their support. The authors thank Drs. No Hee Park, Ki Hyuk Shin, Mo K Kang, and Reuben Kim (UCLA) for sharing their confocal imaging facility. Flow cytometry was performed in the UCLA Jonsson Comprehensive Cancer Center (JCCC) and Center for AIDS Research Flow Cytometry Core Facility that is supported by National Institutes of Health awards P30 CA016042 and 5P30 AI028697, and by the JCCC, the UCLA AIDS Institute, the David Geffen School of Medicine at UCLA, the UCLA Chancellor's Office, and the UCLA Vice Chancellor's Office of Research.

## Additional information

### Competing interests

Sinisa Savic: Received research grant funding and payments from CSL Behring for advisory board participation. The other authors declare that no competing interests exist.

### Funding

| Funder | Grant reference number | Author |
| --- | --- | --- |
| CSL Behring UK | | Laura Rice<br>Sinisa Savic |
| Leeds Hospital Charitable Foundation | | Laura Rice |
| National Institutes of Health | AI083432 | Sonal Srikanth |
| National Institutes of Health | AI083432 | Yousang Gwack |
| National Institutes of Health | AI146615 | Sonal Srikanth |
| National Institutes of Health | AI147063 | Sonal Srikanth |
| National Institutes of Health | AI149236 | Sonal Srikanth |
| National Institutes of Health | AI146615 | Yousang Gwack |
| National Institutes of Health | AI147063 | Yousang Gwack |
| National Institutes of Health | AI149236 | Yousang Gwack |

The funders had no role in study design, data collection and interpretation, or the decision to submit the work for publication.

### Author contributions

Beibei Wu, Data curation, Formal analysis, Methodology; Laura Rice, Clive Carter, Data curation, Formal analysis, Investigation; Jennifer Shrimpton, Kieran Walker, Data curation, Investigation; Dylan Lawless, Formal analysis, Investigation; Lynn McKeown, Formal analysis, Methodology; Rashida Anwar, Formal analysis, Supervision; Gina M Doody, Formal analysis, Methodology, Supervision, Writing – review and editing; Sonal Srikanth, Investigation, Supervision, Writing – original draft, Writing – review and editing; Yousang Gwack, Methodology, Supervision, Writing – original draft, Writing – review and

editing; Sinisa Savic, Conceptualization, Funding acquisition, Investigation, Resources, Supervision, Writing – original draft, Writing – review and editing

**Author ORCIDs**
Beibei Wu ![ORCID] http://orcid.org/0000-0002-0784-0745
Jennifer Shrimpton ![ORCID] http://orcid.org/0000-0002-6346-6516
Dylan Lawless ![ORCID] http://orcid.org/0000-0001-8496-3725
Kieran Walker ![ORCID] http://orcid.org/0000-0002-6694-2979
Sinisa Savic ![ORCID] http://orcid.org/0000-0001-7910-0554

**Ethics**
Human subjects: Ethical approval for this study was obtained from Leeds East Yorkshire Research ethics Committee (18/YH/0070). Written consent to conduct an publish this study was obtained from all participants.

**Decision letter and Author response**
Decision letter https://doi.org/10.7554/eLife.72559.sa1
Author response https://doi.org/10.7554/eLife.72559.sa2

---

# Additional files

**Supplementary files**
• Supplementary file 1. List of all the sgRNAs, and primers used for cDNA sub cloning and qRT-PCR in this study.

• Transparent reporting form

• Source data 1. Source data for all figures.
 Zip folder containing raw and uncropped images for Western blots and Excel spreadsheets of quantitation.

**Data availability**
All data generated or analysed during this study are included in the manuscript and Source data 1. The exception is the complete whole exom sequencing data from the index patient, since this data set contains potentially personally identifiable information and therefore not suitable for unrestricted public access. However, the whole exom sequencing data is available from corresponding author (SiS) on individual request.

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
