## [Editor Report]

The study is the first report and characterization of impaired immune function in an individual with biallelic mutations in the CRACR2A gene. The mechanistic insights from this study, are an important advance in the understanding of CRAC channels and the regulation of calcium dynamics in the T-cell lineage. The work will be of interest to cell biologists, immunologists and those with interests in intracellular signaling.

---

## [Decision Letter]

**Decision letter after peer review:**

Thank you for submitting your article "Biallelic mutations in Calcium Release Activated Channel Regulator 2A (CRACR2A) cause a primary immunodeficiency disorder" for consideration by *eLife*. Your article has been reviewed by 3 peer reviewers, including Apurva Sarin as the Reviewing Editor and Reviewer #1, and the evaluation has been overseen by Satyajit Rath as the Senior Editor. The following individual involved in review of your submission has agreed to reveal their identity: Svetlana Sharapova (Reviewer #3).

Essential revisions:

The study, which is a first such analysis of CRACR2A in a patient, is an important advance understanding of this proteins function in the regulation of CRAC channels in T-cells. The following suggestions are made to address some gaps in information, to strengthen the analysis, and impact of this work.

1. In the experiments shown in Figure 6, control recordings showing calcium entry following calcium readmission to cells exposed to calcium-free solution for the same time (without anti-CD3 or thap), should be included. This signal should be subtracted from the test ones, to provide a better indication of SOCE and know by how much SOCE has been reduced by the CRACR2A mutants.

2. Figure 4: The authors show the calcium signal following crosslinking with anti-CD3 (Figure 4D) but the stimulus is ionomycin/PMA. These calcium signals (ionomycin and PMA) should be shown (healthy control and patient). There does not seem to be any calcium release in Figure 4D in the patient's cells, or the release phase is delayed. It would be helpful to see the responses in calcium-free solution (to monitor release) and then after calcium readmission. And is a 10 minutes recording time frame relevant, when cytokines are measured days later. This is relevant here because the calcium signals between control and patient are identical after 500 seconds. Over longer periods, the overall difference between the two groups (area under curve for example) might be marginal.

3. Table 1: "The patient also had reduced proportion of class switched memory B cells " CD27+ IgM- IgD- (switched memory) percentages, in 2017 – 14.7%, in 2019 – 7.15%, in 2021 – 5.7%, and the normal age-matched control – 5-33%. Could the basis for the conclusion about reduced proportion of class switched memory B cells be clarified?

4. Table 1: In the data on CD25+CD127- (Treg)%, in 2017 – 0,73%, in 2019 – 1.6%, the normal range for Tregs is not provided. The normal range is usually 3-10% if gated on CD4^+^. Could the authors provide an explanation for the Tregs deficiency in peripheral blood?

5. Table 1: Analysis of patients with deficits in STIM and Orai have revealed differences in T-regulatory, TH1 and TH17 subset dependencies on CRAC function (reported in Fuchs et al., J Immunol 2012; Feske et al. Nature 2006; McCarl et al. J. Allergy Clin. Immunol.2009; amongst others). Can the authors comment on these subsets based on the data shown in Table 1? It would be helpful if the authors position the observations made in the patient T-cells, in the context of earlier reports on STIM and Orai loss-of-function analysis in human T-cells.

6. Figure 4: The protocol for Figure 4 may be better explained. In the legend, it is stated: " PBMCs were stimulated with anti-CD3 and anti-CD28 antibodies for 48 hrs and cultured for further 4 days in the presence of IL-2 before re-stimulation with PMA plus ionomycin for 5 hrs for cytokine analysis". Why was such a prolonged stimulation used? Normally, the combination of ionomycin and PMA is sufficient. And the authors are culturing for 4 days with IL-2 and then measuring IL-2 after stimulation with ionomycin/PMA for 5 hours.

7. The KO group is missing in panel 5C.

8. There is a very small difference between the calcium signals (peak and plateau) between KO+ WT and KO+DM (Figure 6B), yet the difference between these two groups on cytokine production (Figure 5) is considerably larger. This could be explained by the fact that the measurements in Figure 6 are bulk calcium recordings, whereas it is local calcium signals near Orai1 channels that drive the cytokine responses in Figure 5 through NFAT activation (e.g. papers by Kar et al.).

9. Immunoprecipitation analysis in Figure 7 shows that CRACR2A can interact with Orai1 and STIM1. Were these experiments done under resting conditions or after stimulation?

10. Have levels of STIM 1 and Orai (Orai1) proteins, been assessed in the patient T-cells? Is STIM1 translocation/puncta formation compromised by the CRACR2A mutants? This can be shown easily by expressing a tagged-STIM1 (e.g. STIM1-YFP).

Additional comments:

1. Could the authors comment on the mechanisms underpinning progressive loss of T-cells observed in the donor? An analysis of T-cell exhaustion markers; CD95 mediated apoptosis or anti-CD3 induced activation induced death in patient cells would be informative. Alternatively, does the expression of CRAC2A mutant proteins in the T-cell line affect the signalling to apoptosis? Along the same lines, can the authors comment if blunted proliferation in response to PMA + Ionomycin (Figure 1D) or x-linking CD3 associated with increased cell death of donor CD4^+^ T-cells?

2. The manuscript would be strengthened by inclusion of direct recordings of SOCE current using the whole cell patch clamp and showing that the CRAC current is reduced when the various CRACR2A mutants are present. However, this is not vital to the conclusions drawn.

3. This is one of the first reports of a SCID associated with aberrant SOCE that does not arise from mutation in either STIM or Orai1 genes. Mutations in STIM1 or Orai1 often evoke non-immunological effects including muscular hypotonia, defective enamel production and anhidrosis, all of which are believed to be a consequence of the loss of SOCE. It would be interesting to know whether the patient in this study exhibited similar phenotypes.

4. Abstract. In the sentence, "The patient exhibited late onset combined immunodeficiency characterised by recurrent chest infections, panhypogammaglobulinemia and CD4^+^ T cell lymphopenia." I would recommend to add current age, gender, ethnicity and his full mutation.

5. P.4 line 22. "He originally presented to gastroenterology services at the age of 19 years with chronic diarrhea". What about chronic diarrhea now? Is there any changes in patient's weight and/height during all period of observation?

6. P.5 line 12-14. "Interestingly, biopsy of his large bowel, which was performed in 2011 for investigation of intermittent diarrhoea, showed mild non-specific inflammatory changes, but also absence of the plasma cells." Any other data from bowel biopsy? T cells? Tregs in biopsy? Information may be included in the text.

7. The discussion may include a comment on what group/table of PID classification this new disease may be included in: Humoral? Combined? Immune-dysregulation? The authors study a lot separate mutations in cell lines, do they have any data from cells obtained from patient's parents?

*Reviewer #1:*

The CRACR2A protein regulates the plasma membrane pore forming unit Orai proteins, which together with the ER membrane localized calcium sensing proteins STIM1/2, is the principal mediator of store operated calcium entry (SOCE) in cells. This study, characterizes T-cell function in an adult who was shown by whole genome sequencing and pedigree analysis to have compound (heterozygous) mutations in both alleles of the CRACR2A gene, and resultant reduced CRACR2A protein expression. Despite the observed deficits in immunoglobin production, the authors establish that B-cell function is largely unimpaired, implicating defects in T-cell function as a defining outcome. This was confirmed by immunological analysis of the affected individual's T-cells. The authors employ multiple approaches to characterize the effects of the mutations, including the over-expression of mutant proteins in T-cells with ablations of endogenous CRACR2A protein. Mechanistic insights are provided by a combination of cell biological, biochemical and functional analysis. Together, these identify molecular features underpinning regulation of calcium entry and activation of the MAPK pathway by CRAC2A in T-cells. Although the analysis is based on one patient, key observations are reproduced by recombinant protein expression, providing important mechanistic insights on CRACR2A function T-cells.

The analysis of CRACR2A in T-cell subsets, which are known to differ in their dependence on CRAC channel function are however, not addressed in this study.

*Reviewer #2:*

This study by Wu and colleagues reports the presence of compound heterozygous mutations in CRACR2A and dissects out how these cause a primary immunodeficiency in a patient of East Asian origin. This is one of the first reports of a SCID associated with aberrant SOCE that does not arise from mutation in either STIM or Orai1 genes. The authors show that the clinical characterization is a decrease in antibody production, due to lymphopenia and impaired T cell activation. Wu et al. further demonstrate that the latter arises through a reduction in store-operated calcium entry (SOCE), JNK phosphorylation and a decrease in IFN-γ, IL-2 and TNF expression. The findings are novel and interesting and the paper is well-written and easy to follow. The data are convincing and adequately controlled. I support publication. Nevertheless, there are a few issues that the authors might wish to consider.

---

## [Author Response]

Essential revisions:The study, which is a first such analysis of CRACR2A in a patient, is an important advance understanding of this proteins function in the regulation of CRAC channels in T-cells. The following suggestions are made to address some gaps in information, to strengthen the analysis, and impact of this work.1. In the experiments shown in Figure 6, control recordings showing calcium entry following calcium readmission to cells exposed to calcium-free solution for the same time (without anti-CD3 or thap), should be included. This signal should be subtracted from the test ones, to provide a better indication of SOCE and know by how much SOCE has been reduced by the CRACR2A mutants.

Thank you for pointing this out. Based on our more than a decade of experience in SOCE measurements with primary human/mouse T cells and cell lines including Jurkat T cells, we have observed minimal changes in Ca^2+^ entry upon exchange of 0 or 2- mM Ca^2+^-containing solution, unless there is a stimulus present (like thapsigargin or anti-CD3 ab crosslinking or a Ca^2+^-dependent mutant expression in the cells). Accordingly, when we present the bar graphs (Figures 4D/E and 6A/B), we subtract the baseline (average readings in 2 mM Ca^2+^-containing solution, prior to addition of any stimulus) and present the data as “Peak-basal (ratio)”. We include a figure from one of our papers (Author response image 1, from Figure 2 of PMID: 22586105) where we show the same in primary murine T cells. When we compare steady state ratios at the beginning of the experiment in 2 mM Ca^2+^-containing solution (time 100 s) and the end after exchange with Ca^2+^ free solution (time 900 s), there is minimal change in ratio values, as seen by the black line. Our observations are similar with Jurkat T cells as well.

**Author response image 1. sa2fig1:** Levels of intracellular Ca^2+^ are measured in primary T cells by exchanging Ca^2+^-free solution with that containing 2 mM CaCl_2_. Copied from PMID: 22586105.

2. Figure 4: The authors show the calcium signal following crosslinking with anti-CD3 (Figure 4D) but the stimulus is ionomycin/PMA. These calcium signals (ionomycin and PMA) should be shown (healthy control and patient). There does not seem to be any calcium release in Figure 4D in the patient's cells, or the release phase is delayed. It would be helpful to see the responses in calcium-free solution (to monitor release) and then after calcium readmission. And is a 10 minutes recording time frame relevant, when cytokines are measured days later. This is relevant here because the calcium signals between control and patient are identical after 500 seconds. Over longer periods, the overall difference between the two groups (area under curve for example) might be marginal.

We repeated some of the experiments where we first stimulate SOCE by anti-CD3 cross linking and subsequently added ionomycin to see maximal levels of SOCE in control and patient cells. The new data presented in Figure 4D of the revised manuscript shows reduced peak as well as sustained levels after anti-CD3 antibody treatment in patient cells. Further this reduction is sustained even after ionomycin treatment. These data were further verified by measuring SOCE using a different agonist – thapsigargin, which inhibits the SERCA pump to passively deplete the ER Ca^2+^ stores and thereby activate SOCE. As shown in representative traces in Figure 4E of the revised manuscript, the patient cells show similar levels of ER Ca^2+^ stores (area under curve in the presence of thapsigargin in Ca^2+^-free solution), however SOCE is reduced after re-introduction of 2 mM Ca^2+^-containing Ringers solution. Further, in our previous studies, we have shown that CRACR2A is important for both Ca^2+^ entry triggered by anti-CD3 Ab cross-linking and passive ER Ca^2+^ store depletion by treatment with thapsigargin or ionomycin (PMID: 27016526). In this manuscript, we also show these results in Figure 6A and B, using Jurkat T cells deleted for endogenous CRACR2A. Taken together, using either thapsigargin, anti-CD3-crosslinking or ionomycin stimuli, we can see reduced SOCE in either patient cells or Jurkat T cells deleted for endogenous CRACR2A.

3. Table 1: "The patient also had reduced proportion of class switched memory B cells " CD27+ IgM- IgD- (switched memory) percentages, in 2017 – 14.7%, in 2019 – 7.15%, in 2021 – 5.7%, and the normal age-matched control – 5-33%. Could the basis for the conclusion about reduced proportion of class switched memory B cells be clarified?

The explanation for reduced memory B cells is likely to be the same as for panhypogammaglobulinemia, which is due to inadequate T cell help. We have added a sentence in the discussion (Page 9) to clarify this point: “The same mechanism is likely to cause reduction in proportion of class switched memory B cells.”

4. Table 1: In the data on CD25+CD127- (Treg)%, in 2017 – 0,73%, in 2019 – 1.6%, the normal range for Tregs is not provided. The normal range is usually 3-10% if gated on CD4^+^. Could the authors provide an explanation for the Tregs deficiency in peripheral blood?

We have repeated T regulatory cells evaluation recently, and also included results from additional evaluation from an earlier time point in 2021. These have now been included into the Table 1. The evaluation in February of 2021 showed % of T regs to be 2.9 whilst the repeat measurement in November showed % of Tregs to be 3.5, which is just above the lower end of the normal reference range provided by the reviewer. Overall, the patient has detectable T regs, albeit at the lower end or just below the normal range. One likely explanation for this observation is the patient’s poor IL-2 response/production, which is particularly important for Treg development and maintenance. (reference: Setoguchi R, Hori S, Takahashi T, Sakaguchi S. Homeostatic maintenance of natural Foxp3+CD25+CD4^+^ regulatory T cells by interleukin (IL)-2 and induction of autoimmune disease by IL-2 neutralization. J Exp Med 2005; 201: 723– 35.). However, considering the lack of any autoimmune/inflammatory or lymphoproliferative complications, the apparent partial reduction in T regs is unlikely to be clinically relevant in this case.

5. Table 1: Analysis of patients with deficits in STIM and Orai have revealed differences in T-regulatory, TH1 and TH17 subset dependencies on CRAC function (reported in Fuchs et al., J Immunol 2012; Feske et al. Nature 2006; McCarl et al. J. Allergy Clin. Immunol.2009; amongst others). Can the authors comment on these subsets based on the data shown in Table 1? It would be helpful if the authors position the observations made in the patient T-cells, in the context of earlier reports on STIM and Orai loss-of-function analysis in human T-cells.

In the case of our CRACR2A deficient patient we have shown that patient’s T cells have significantly reduced capacity to produce IFNγ, IL-2 and TNF when compared to healthy controls (Figure 4A-C). This is similar to what has previously been shown in patients with STIM1 and ORA1 deficiencies. We did not specifically investigate IL-4 or IL-17 production by the T cells. Nevertheless, we performed additional phenotyping experiments to determine the ratio of TH1/TH2 and TH17 T cells in the peripheral blood. We did not find any significant differences between the patient and healthy controls, although the proportion of the patient derived TH2 cells seemed reduced in comparison with controls (see Suppl Figure 1). However, the method we used to determine the TH1/TH2 and TH17 ratios in the peripheral blood relies on the expression of surface chemokine receptors rather than cytokine production, therefore functionally these cells might still be impaired, as we have demonstrated for TH1 IFNγ producing subset.

6. Figure 4: The protocol for Figure 4 may be better explained. In the legend, it is stated: " PBMCs were stimulated with anti-CD3 and anti-CD28 antibodies for 48 hrs and cultured for further 4 days in the presence of IL-2 before re-stimulation with PMA plus ionomycin for 5 hrs for cytokine analysis". Why was such a prolonged stimulation used? Normally, the combination of ionomycin and PMA is sufficient. And the authors are culturing for 4 days with IL-2 and then measuring IL-2 after stimulation with ionomycin/PMA for 5 hours.

PBMCs were stimulated with anti-CD3 and anti-CD28 antibodies to induce their differentiation and proliferation. After first 48 hours of stimulation, cells were detached from the antibody-coated plates and expanded with IL-2. After 4 days of expansion, effector T cells were acutely re-stimulated with PMA and ionomycin treatment for 5 h to check cytokine levels. Since we wash away IL-2 before re-simulation and check IL-2 using intracellular staining, addition of IL-2 during the expansion does not influence data interpretation. These methods have been better explained in the methods section of the revised manuscript.

7. The KO group is missing in panel 5C.

Panel 5C examines a possible dominant negative effect of mutants of CRACR2A in the presence of endogenous WT CRACR2A protein. Hence, these experiments cannot be done in KO cells. Reconstitution efficiency of mutant CRACR2A proteins is shown in KO cells in Figures 5A and 5B.

8. There is a very small difference between the calcium signals (peak and plateau) between KO+ WT and KO+DM (Figure 6B), yet the difference between these two groups on cytokine production (Figure 5) is considerably larger. This could be explained by the fact that the measurements in Figure 6 are bulk calcium recordings, whereas it is local calcium signals near Orai1 channels that drive the cytokine responses in Figure 5 through NFAT activation (e.g. papers by Kar et al.).

Thank You for pointing this out. Of course, one of the mechanisms can be local Ca^2+^ concentration in microdomains of CRAC channels that mediate NFAT translocation. We will need to use advanced microscopy techniques including TIRF microscopy to ascertain that. This aspect will form a part of future studies of this project.

9. Immunoprecipitation analysis in Figure 7 shows that CRACR2A can interact with Orai1 and STIM1. Were these experiments done under resting conditions or after stimulation?

Immunoprecipitation experiments were done purely to check binding between the proteins when overexpressed in HEK293T cells, hence there was no stimulation involved.

10. Have levels of STIM 1 and Orai (Orai1) proteins, been assessed in the patient T-cells? Is STIM1 translocation/puncta formation compromised by the CRACR2A mutants? This can be shown easily by expressing a tagged-STIM1 (e.g. STIM1-YFP).

In the revised manuscript, we include experiments to check protein and transcripts of ORAI1 and STIM1 in patient cells. As seen in newly added supplementary figure 2, protein and transcript expression of *ORAI1* and *STIM1* is similar in patient cells as compared to representative healthy controls. As previously published, overexpression of tagged WT STIM1 can often rescue the defects observed in CRACR2A KO cells, due to compensatory effects, hence we cannot use that system for detection of STIM1 translocation to puncta in patient cells.

Additional comments:1. Could the authors comment on the mechanisms underpinning progressive loss of T-cells observed in the donor? An analysis of T-cell exhaustion markers; CD95 mediated apoptosis or anti-CD3 induced activation induced death in patient cells would be informative. Alternatively, does the expression of CRAC2A mutant proteins in the T-cell line affect the signalling to apoptosis? Along the same lines, can the authors comment if blunted proliferation in response to PMA + Ionomycin (Figure 1D) or x-linking CD3 associated with increased cell death of donor CD4^+^ T-cells?

Thank You for providing an interesting suggestion. This finding is unique for human T cells as *Cracr2a^fl/fl^ Cd4*-cre mice do not show significant reduction in CD4^+^ cell numbers (Woo JS et al., Journal of Immunology, 2018, PMID 29987160). However, while humans are constantly exposed to pathogens and diverse microbiota, *Cracr2a^fl/fl^ Cd4*-cre mice are housed in a specific pathogen-free facility with limited microbiota, suggesting that the reduced number of CD4^+^ cells in P1 may be indeed derived from defects in TCR signaling, which plays a crucial role in T cell homeostasis, survival, and death. Although we completely agree with the reviewer’s opinion, it is currently hard to check this idea due to limited availability of patient cells. In the future, we plan to check detailed T cell responses to cell death using knockout mice after in vivo stimulation with anti-CD3 antibody injection by checking CD95 and other extrinsic and intrinsic T cell death mechanisms.

2. The manuscript would be strengthened by inclusion of direct recordings of SOCE current using the whole cell patch clamp and showing that the CRAC current is reduced when the various CRACR2A mutants are present. However, this is not vital to the conclusions drawn.

Thank You for this suggestion. It is technically very challenging to measure endogenous CRAC currents in wild type Jurkat T cells. Deciphering differences in current densities in CRACR2A KO Jurkat T cells expressing various mutants of CRACR2A may not yield significant results, mainly due to very low signal to noise ratio in those measurements.

3. This is one of the first reports of a SCID associated with aberrant SOCE that does not arise from mutation in either STIM or Orai1 genes. Mutations in STIM1 or Orai1 often evoke non-immunological effects including muscular hypotonia, defective enamel production and anhidrosis, all of which are believed to be a consequence of the loss of SOCE. It would be interesting to know whether the patient in this study exhibited similar phenotypes.

The patient did not exhibit any non-immunological features that are typically seen in STIM1 or ORAI1 deficiencies. This information has now been added to the results (page 4) of the revised manuscript.

4. Abstract. In the sentence, "The patient exhibited late onset combined immunodeficiency characterised by recurrent chest infections, panhypogammaglobulinemia and CD4^+^ T cell lymphopenia." I would recommend to add current age, gender, ethnicity and his full mutation.

These details have now been added to the abstract.

5. P.4 line 22. "He originally presented to gastroenterology services at the age of 19 years with chronic diarrhea". What about chronic diarrhea now? Is there any changes in patient's weight and/height during all period of observation?

Chronic diarrhea resolved after the patient was commenced on immunoglobulin replacement therapy. We did not find an obvious explanation for his diarrhea. The diarrhea could have been a consequence of immuno-dysregulation resulting from the profound panhypogammaglobulinemia. This phenomenon is observed frequently in clinical practice, whereby different inflammatory and autoimmune complications, including inflammatory bowel manifestations are corrected following institution of the immunoglobulin replacement. We have added a sentence in the results (page 5) of the revised manuscript to update the patient status related to the chronic diarrhoea: “His chronic diarrhoea resolved following commencement of IVIG and the ongoing treatment with IVIG has resulted in excellent control of respiratory infections.”

6. P.5 line 12-14. "Interestingly, biopsy of his large bowel, which was performed in 2011 for investigation of intermittent diarrhoea, showed mild non-specific inflammatory changes, but also absence of the plasma cells." Any other data from bowel biopsy? T cells? Tregs in biopsy? Information may be included in the text.

There was no further information from the biopsy.

7. The discussion may include a comment on what group/table of PID classification this new disease may be included in: Humoral? Combined? Immune-dysregulation? The authors study a lot separate mutations in cell lines, do they have any data from cells obtained from patient's parents?

Regarding the classification we added a following sentence at end of the Discussion section:” Given the resultant clinical phenotype and range of associated immunological abnormalities, we propose that CRACR2A deficiency should be included within the group of combined immunodeficiency disorders in the IEI classification”.

Unfortunately, we only had the DNA available from the parents.